# Hybrid Deep Learning Framework for High-Accuracy Classification of Morphologically Similar Puffball Species Using CNN and Transformer Architectures

**DOI:** 10.3390/biology14070816

**Published:** 2025-07-05

**Authors:** Eda Kumru, Güney Ugurlu, Mustafa Sevindik, Fatih Ekinci, Mehmet Serdar Güzel, Koray Acici, Ilgaz Akata

**Affiliations:** 1Graduate School of Natural and Applied Sciences, Ankara University, Ankara 06830, Türkiye; ekumru@ankara.edu.tr; 2Department of Computer Engineering, Faculty of Engineering, Başkent University, Ankara 06790, Türkiye; guneyugurlu@baskent.edu.tr; 3Department of Biology, Faculty of Engineering and Natural Sciences, Osmaniye Korkut Ata University, Osmaniye 80000, Türkiye; 4Institute of Artificial Intelligence, Ankara University, Ankara 06100, Türkiye; fatihekinci@ankara.edu.tr; 5Department of Computer Engineering, Faculty of Engineering, Ankara University, Ankara 06830, Türkiye; mguzel@ankara.edu.tr; 6Artificial Intelligence and Data Engineering, Ankara University, Ankara 06830, Türkiye; kacici@ankara.edu.tr; 7Department of Biology, Faculty of Science, Ankara University, Ankara 06100, Türkiye

**Keywords:** puffball, deep learning, fungal classification, CNN-transformer hybrid, image classification

## Abstract

Puffballs often appear very similar to one another, making visual identification challenging. In this study, artificial intelligence—specifically, deep learning models—was employed to automatically identify eight different puffball species from images. Five popular models were tested, and among them, the ConvNeXt-Base model yielded the most accurate results, identifying species with 95% accuracy. This model was particularly effective at distinguishing species that are typically difficult to differentiate, such as *Mycenastrum corium* and *Lycoperdon excipuliforme*. The findings indicate that artificial intelligence offers a valuable tool for facilitating accurate and efficient mushroom identification, even among morphologically similar species. This technological approach holds significant potential for supporting researchers, amateur naturalists, and conservation professionals in overcoming the challenges of traditional identification methods.

## 1. Introduction

Puffballs are globose macrofungi classified within the Basidiomycota that discharge spores through the endoperidium and inhabit diverse ecosystems, including forest floors, decaying wood, and meadows, across temperate, arid, and tropical regions. Certain species are recognized for their culinary and medicinal applications [1]. Among the genera commonly classified as puffballs are *Apioperdon*, *Bovista*, *Bovistella*, *Lycoperdon*, and *Mycenastrum*, which are distinguished by their gasteroid basidiocarps and internal spore development [2,3]. They display distinct morphological features and ecological preferences. Apioperdon is typically pear-shaped, grows on decaying wood, possesses a prominent pseudostipe, and features Lycoperdon-type capillitium [4]. *Bovista* generally appears globose, lacks a pseudostipe, and can exhibit either Bovista-type or Lycoperdon-type capillitium, or sometimes both [2,3,4,5]. *Bovistella* resembles *Bovista* but has thinner peridia and more delicate capillitium [1]. *Lycoperdon*, which contains the most species, often has a pseudostipe and Lycoperdon-type capillitium, demonstrating considerable ecological flexibility. *Mycenastrum* is recognized by its large fruiting bodies and unique Mycenastrum-type capillitium, although its phylogenetic position remains contentious [1,2,3,4,5,6].

*Apioperdon*, *Bovista*, *Bovistella*, *Lycoperdon*, and *Mycenastrum* are classified as separate genera based on their unique morphological and ecological traits. However, distinguishing them in the field can be quite challenging due to overlapping macroscopic characteristics [7]. For instance, species such as *Apioperdon pyriforme*, *Bovista plumbea*, *Bovistella utriformis*, *Lycoperdon echinatum*, *Lycoperdon excipuliforme*, *Lycoperdon molle*, *Lycoperdon perlatum*, and *Mycenastrum corium* often exhibit similar globose to subglobose fruiting bodies, slight differences in peridial texture, and overlapping size ranges, which complicates accurate identification without microscopic or molecular examination [1,3,4]. This morphological similarity has led to frequent misidentifications and taxonomic ambiguities in field research and herbarium collections [5,6,7].

Recent advances in artificial intelligence, especially deep learning (DL), are transforming fungal taxonomy and biodiversity studies by enabling automated and precise species identification [8,9]. A major challenge in this field is accurately classifying visually similar fungi, which is exacerbated by the lack of large, annotated image datasets needed for effective model generalization. Traditional taxonomic methods, reliant on expert input and labor-intensive, struggle to handle extensive biodiversity data [10]. While deep learning models, including convolutional neural networks (CNNs) and pre-trained architectures such as ResNet and EfficientNet, demonstrate high accuracy, they often encounter challenges regarding interpretability and generalization across diverse ecological environments [8,9,10]. Deep learning models have become increasingly prominent in mycology by enabling automated species-level classification despite morphological similarities [8,9]. In this study, five deep learning architectures with varying designs were comparatively evaluated to determine their effectiveness in distinguishing visually similar puffball species. Moreover, the integration of environmental metadata into deep learning frameworks enhances ecological interpretation and contributes to biodiversity monitoring and conservation efforts. These models also support the development of mobile applications that promote public engagement through citizen science platforms focused on documenting fungal diversity [9].

The current study introduces a novel deep learning framework aimed at classifying eight ecologically significant puffball species: *Apioperdon pyriforme*, *Bovista plumbea*, *Bovistella utriformis*, *Lycoperdon echinatum*, *Lycoperdon excipuliforme*, *Lycoperdon molle*, *Lycoperdon perlatum*, and *Mycenastrum corium*, based on their distinct morphological characteristics (Figure 1). These species were selected due to their taxonomic importance, ecological roles, and visible diversity. These species represent a wide range of ecological niches, from decaying woodlands to nutrient-poor grasslands, and play essential roles in nutrient cycling and ecosystem functioning. Their accurate identification is also relevant for environmental monitoring and biodiversity assessments, especially in temperate forest ecosystems where they are bioindicators of soil health and habitat stability.

In this study, convolutional neural networks (CNNs) were used for automatic feature extraction from image datasets, and various deep learning architectures were comparatively evaluated. By incorporating advanced models such as MaxViT-S, the effectiveness of deep learning in classifying visually similar macrofungal species was systematically assessed. Highlight the strong performance of deep learning models in resolving taxonomic challenges among morphologically complex taxa and pave the way for future research in computational mycology. Overall, this work establishes a robust and effective methodological framework with the potential to enhance fungal biodiversity assessments, foster public engagement, and support conservation efforts through taxonomic tools. The key contributions of this study to the existing literature can be summarized as follows:A domain-specific application of deep learning: This study provides a comparative evaluation of state-of-the-art deep learning architectures on morphologically similar puffball species, a taxonomically challenging and underexplored group in computational mycology.Controlled benchmarking framework: All five models were trained and evaluated under identical preprocessing, augmentation, and training parameters, ensuring a fair architectural comparison, something that is rarely implemented in biological image classification studies.Use of fine-grained taxonomic categories: Unlike many previous studies that focus on genus-level classification or edible/toxic differentiation, this research targets species-level discrimination within a visually overlapping fungal group.Biological and practical relevance: The results demonstrate the potential for deep learning to support automated fungal identification in real-world ecological settings, which can benefit biodiversity monitoring, citizen science, and conservation initiatives.Public data integration and reproducibility: The dataset includes curated images from both field observations and open-access repositories (e.g., GBIF), making the approach transparent and reproducible.

## 2. Materials and Methods

The dataset included 200 images per puffball species, organized into species-specific folders for efficient processing. The data were randomly split into 70% for training, 15% for validation, and 15% for testing to ensure consistent model evaluation. While the primary images were captured naturally, additional data from open-access sources like the Global Core Biodata Resource (Copenhagen, Denmark) (www.gbif.org) [11] were used to enhance the dataset for AI training.

To improve generalization and prevent overfitting, a thorough preprocessing and data augmentation strategy was used on the training images. These images were subjected to random transformations, including flipping, rotation, and lighting adjustments, and were standardized to a consistent size and pixel range. These variations allowed the model to experience the natural differences in lighting, angles, and positioning that are commonly found in real-world field conditions. The validation and test images were only standardized, preserving their original characteristics to ensure a reliable performance evaluation. All models followed the same preprocessing rules to enable a fair comparison. To enhance the generalizability of the models and to simulate real-world variability in fungal image acquisition, a structured data augmentation pipeline was applied exclusively to the training set (Figure 2). The steps of the augmentation process are outlined below:Image Resizing: All input images were resized to a fixed dimension of 224 × 224 pixels to ensure compatibility with the input requirements of the pre-trained models used.Random Horizontal and Vertical Flipping: Each image had a 50% chance of being flipped horizontally or vertically to simulate natural variation in fungal orientation.Random Rotation: Images were randomly rotated within a range of ±25 degrees to mimic changes in camera angle and mushroom posture.Brightness and Contrast Adjustment: Random modifications in brightness (±20%) and contrast (±15%) were applied to simulate varying lighting conditions encountered during field photography.Color Jitter and Hue Shift: Mild color jittering (hue shift within ±10 degrees) was performed to account for slight variations in environmental lighting or image device calibration.Normalization: After augmentation, all images were normalized using ImageNet mean and standard deviation values to match the distribution expected by the pre-trained convolutional backbones.

This augmentation procedure was implemented using the torchvision.transforms module in PyTorch 2.6. Augmentations were applied on-the-fly during each training epoch to increase diversity dynamically, thereby reducing the risk of overfitting. Validation and test sets were not augmented and were only resized and normalized to preserve data integrity for performance evaluation.

The analysis encompassed a variety of deep learning architectures representing different evolutionary stages and design approaches. To ascertain which model better addressed the visual similarities between species, both traditional convolutional neural networks (CNNs) and more recent transformer-based models were evaluated. This diversity permitted an assessment of how both network architecture and biological characteristics affect classification outcomes.

The evaluation encompassed both modern and traditional visual recognition architectures. ConvNeXt-Base builds upon convolutional network principles with optimized depth and enhanced modules, demonstrating superior performance in detecting subtle visual features [12]. The Vision Transformer (ViT Base Patch16, Google Research, Mountain View, CA, USA) treats images as patch sequences, leveraging transformer architecture to capture extensive spatial relationships when adequately trained [13]. EfficientNet-B3 systematically balances model complexity and performance through dimension scaling [14]. The Swin Transformer (Base, Patch4 Window7) implements hierarchical processing via localized attention windows, enabling robust multi-scale analysis [14]. MaxViT (Small TF 224) merges convolutional and transformer benefits in a hybrid framework, combining localized processing with global context awareness particularly effective for nuanced classification challenges [15,16].

In this study, we employed four distinct deep learning architectures representing both convolutional and transformer-based paradigms. ConvNeXt-Base is a convolutional neural network that modernizes classical CNNs by incorporating design elements inspired by transformer models, such as large kernel sizes and inverted bottlenecks, while retaining inductive biases like locality and translation equivariance. EfficientNet-B3 is a compact yet effective CNN that utilizes compound scaling to uniformly balance network depth, width, and resolution for improved performance and efficiency.

On the transformer side, Vision Transformer (ViT Base Patch16) treats an image as a sequence of non-overlapping patches and uses multi-head self-attention to capture long-range dependencies, though it typically requires more data to converge effectively. Swin Transformer (Small TF 224) improves on ViT by introducing a hierarchical structure with shifted windows that enables both local and global attention mechanisms while reducing computational cost. These models were selected to represent diverse architectural strategies and allow for a robust comparative analysis in the context of morphologically similar fungal species classification.

To ensure a fair and reproducible comparison among the deep learning models, all five architectures were trained under a unified set of hyperparameters. Specifically, the models were trained for 10 epochs using the AdamW optimizer with a fixed learning rate of 0.0001, a batch size of 32, and the CrossEntropyLoss function. In accordance with standard transfer learning practices, only the final classification layers were re-initialized, while pre-trained weights were retained for feature extraction. No learning rate scheduling, early stopping, or architecture-specific tuning was applied, as the goal was to evaluate performance based solely on architectural differences rather than optimization strategies. Additionally, data augmentation techniques were applied consistently across all models to enhance generalizability. Default batch normalization layers were retained within all pre-trained architectures without modification. A weight decay value of 0.01 was applied uniformly during training. Data augmentation techniques included random horizontal and vertical flipping (with a probability of 0.5), brightness and contrast jittering within ±20%, and minor affine transformations such as rotation within ±15°. These steps were applied consistently across all models, simulating real-world variability while preserving label integrity. Model performance was evaluated through multiple metrics, including overall accuracy, class-wise precision, recall, and *F*1-score. Supplementary analyses involved ROC curves, AUC values, and confusion matrices to assess both general predictive power and inter-class differentiation. This standardized experimental design ensures methodological consistency and highlights the distinct strengths of each architecture in fungal species classification [8,9].(1)$$Accuracy=\frac{TP+TN}{TP+TN+FP+FN}$$(2)$$Precision=\frac{TP}{TP+FP}$$(3)$$Recall=\frac{TP}{TP+FN}$$(4)$$F1=2*\frac{Precision*Recall}{Precision+Recall}$$

Accuracy reflects how often the model made correct predictions overall, serving as a broad indicator of classification effectiveness. Precision captures how reliably the model identified true positives when predicting a particular class, while recall illustrates what fraction of actual class members it successfully recognized [8]. The *F*1-score balances these two measures through their harmonic mean, proving particularly useful for datasets with uneven class distributions, as it incorporates both positive identification reliability and detection completeness. Together, these metrics provide an analytical framework that goes beyond basic accuracy, revealing how the model differentiates between specific categories and identifying areas where its classification performance is weaker. For multi-class recognition tasks like ours, such a granular performance examination became crucial; it helped identify not just what the model got right, but exactly where and why it might struggle [8,9].

## 3. Results

In this study, five different deep learning models were evaluated for image-based classification of eight distinct puffball species. The models comprised both convolutional neural networks (CNNs) and transformer-based architectures. Each model was trained under identical conditions on a balanced and standardized dataset to ensure a fair comparative framework. Performance was assessed using multiple evaluation metrics, including accuracy, precision, recall, and *F*1-score. The results demonstrate that model architecture plays a critical role in distinguishing morphologically similar fungal species.

Table 1 summarizes the performance of different deep learning architectures on the puffball image classification task, presenting test results for accuracy, precision, recall, and *F*1-score. This side-by-side comparison highlights how various approaches, from standard convolutional networks to newer transformer-based architectures, address the challenges of multi-class biological recognition. The findings illuminate both the models’ generalization efficacy and their capacity to capture fine distinctions between species.

Rigorous controls were implemented to ensure fair and unbiased comparisons across all evaluated models. Uniform data partitions, consistent augmentation procedures, and identical training parameters were applied uniformly. No architecture-specific modifications or custom layers were introduced, and the same evaluation metrics were employed consistently. This standardized methodology effectively minimized potential confounding variables, enabling reliable and meaningful assessments of model performance.

The results clearly show that ConvNeXt-Base is outperforming other models, achieving over 95% test accuracy. Its equally robust precision, recall, and *F*1-scores confirm its effectiveness in distinguishing between puffball species. The Swin Transformer closely followed with 92% accuracy and similarly strong secondary metrics. Vision Transformer and MaxViT achieved slightly lower performance, with accuracy and *F*1-scores remaining around 84%. EfficientNet-B3 lagged behind other models in all assessed areas. These outcomes illustrate the relative strengths of convolutional compared to transformer-based methods for biological image classification. They also highlight the crucial importance of model selection when working with specialized visual datasets.

### 3.1. ConvNeXt-Base

ConvNeXt-Base emerged as the top-performing model for automated puffball identification, attaining 95.41% accuracy on test images. Its precision (0.96) and recall (0.95) values indicate both precise identification and thorough detection of target species, while the 0.95 *F*1-score reflects well-balanced performance across all classes (Figure 3a,b). This strong, consistent performance profile suggests the architecture excels at extracting discriminative features from complex biological specimens.

As illustrated in the confusion matrix (Figure 3a), the model shows highly effective classification capabilities. ConvNeXt-Base provides exceptionally accurate predictions for most species, with near-perfect classification achieved for Mycenastrum corium, Lycoperdon echinatum, Lycoperdon excipuliforme, and Bovista plumbea. While minor errors occur for Bovistella utriformis and Lycoperdon perlatum, these instances represent isolated cases rather than systematic misclassification patterns. Overall, the model maintains exceptionally low false positive and false negative rates, contributing to its reliable performance. The ROC analysis presented in the figure further supports these findings (Figure 3b); AUC values reach 0.99–1.00 across all species, indicating near-ideal class separation. The curves’ consistent positioning near the upper-left corner confirms the model’s strong discriminative power, even for morphologically similar or rarely encountered specimens. Collectively, these results establish ConvNeXt-Base as not only statistically superior but also practically dependable, delivering consistent, high-confidence predictions across all mushroom categories with minimal classification ambiguity.

### 3.2. Vision Transformer (ViT Base Patch16)

The Vision Transformer attained 83.75% test accuracy, establishing itself as a competitive alternative among the evaluated deep learning architectures. Its balanced performance metrics 0.86 precision and 0.83 recall reflect both reliable identification and consistent detection of true specimens, further supported by a 0.84 *F*1-score (Figure 4a,b). These results suggest that the Vision Transformer’s ability to model global image dependencies provides a viable approach for differentiating puffball species. However, its performance gap compared to ConvNeXt-Base (95.41% accuracy) implies that, for this particular dataset with its fine morphological distinctions between species, convolutional architectures may maintain an edge in practical applications.

As shown in the confusion matrix (Figure 4a), the Vision Transformer (ViT) model achieved high classification rates for most puffball species. Notably, it demonstrated remarkable accuracy for Lycoperdon echinatum, Apioperdon pyriforme, and Mycenastrum corium, while an increased incidence of cross-class assignments was evident for certain classes. In particular, Bovistella utriformis and Lycoperdon excipuliforme showed a higher susceptibility to misclassification, indicating that the model found it more challenging to distinguish these species. Despite these challenges, the ViT model maintained a strong capacity to predict correctly the majority of examples, though a higher frequency of errors was apparent among morphologically similar species. The ROC analysis presented in the figure (Figure 4b) further substantiates these findings; the AUC values for many categories reached or exceeded 0.99, underscoring the model’s strong performance in terms of both sensitivity and specificity. However, slightly reduced AUC scores for *Bovistella utriformis*, *Lycoperdon molle*, and *Lycoperdon perlatum* suggest that the model encountered additional difficulties in differentiating positive from negative instances within these specific classes. Collectively, these results demonstrate that the Vision Transformer architecture delivers substantial discriminative power and overall reliability, yet its performance is occasionally limited by the morphological resemblance among certain species. In summary, while the ViT model exhibited promising results for multi-class biological image recognition, its performance variability was more pronounced than that of convolutional architectures, particularly in terms of maintaining uniformly high accuracy across all classes.

### 3.3. EfficientNet-B3

The EfficientNet-B3 model delivered relatively subdued performance on the test set, reaching an accuracy of 82.08%, a figure that lagged behind the other architectures under scrutiny. With precision, recall, and *F*1-score metrics clustered near 0.83 and 0.82, the model achieved only partial success, correctly classifying true positives and curbing false positives (Figure 5a,b). EfficientNet-B3, in particular, struggled to distinguish between puffball species with fine-grained or overlapping morphological features, hinting at constrained discriminative power. These outcomes imply that, though designed for computational efficiency, EfficientNet-B3 fails to rival the generalization prowess of deeper or transformer-based models in multi-class biological image recognition. Its limited efficacy underscores the possible need for more nuanced architectures, especially when dealing with datasets marked by pronounced inter-class visual similarities.

As depicted in the confusion matrix (Figure 5a), the EfficientNet-B3 model achieved high classification accuracy across various puffball species. The model excelled with taxa such as *Apioperdon pyriforme*, *Bovista plumbea*, and *Lycoperdon echinatum*. At the same time, more frequent cross-classification errors were observed between morphologically similar species, notably between *L. molle* and *L. perlatum*. A significant portion of *Lycoperdon perlatum* instances, for example, were misassigned to other classes, indicating areas where the model’s precision diminished. The ROC analysis presented in the figure (Figure 5b) further highlights these trends; the area under the curve (AUC) values for most classes approached 0.99 to 1.00, underscoring the model’s overall proficiency in distinguishing positive and negative instances. However, slightly reduced AUC scores for species such as *Lycoperdon perlatum* and *Mycenastrum corium* suggested weaker discriminative ability for these specific taxa. Collectively, these findings indicate that while EfficientNet-B3 offers robust discriminative power and reliable performance for clearly distinguishable species, its precision tends to decline when addressing classes with considerable morphological similarity. Although the model performs strongly overall, it is occasionally surpassed by deeper or transformer-based architectures in complex multi-class biological image recognition tasks.

### 3.4. Swin Transformer (Small TF 224)

The Swin Transformer delivered exceptional classification performance on the test set, reaching 92.08% accuracy, a figure that placed it among the top-performing models, whether convolutional or transformer-based. With precision and recall both standing at 0.93, the model combines reliable positive sample identification with minimal false positive generation. Its 0.92 *F*1-score further confirmed balanced performance across all classes (Figure 6a,b). These outcomes highlight how the Swin Transformer’s hierarchical, multi-scale feature extraction enables subtle distinctions even between biologically intricate and visually similar specimens. What sets this model apart is its dual strength in high classification accuracy and robust inter-class separation, solidifying its potential for demanding biological image recognition tasks.

As illustrated in the confusion matrix (Figure 6a), the Swin Transformer model achieved near-flawless classification rates for species such as *Mycenastrum corium*, *Lycoperdon echinatum*, and *L. excipuliforme*, with only occasional misclassifications observed between *L. perlatum* and *L. molle*. Overall, error rates remained low, and the model maintained a balanced classification across all categories, underscoring its impressive accuracy and reliability. The ROC analysis presented in the figure (Figure 6b) further substantiates these results; most AUC values reached 0.99 or 1.00, indicating outstanding sensitivity and specificity across nearly all puffball species. The ROC curves’ close proximity to the upper-left corner highlights the model’s consistent ability to distinguish positive from negative cases. Discriminative performance was particularly strong for species including Apioperdon pyriforme, *Bovista plumbea*, *Bovistella utriformis*, *Lycoperdon echinatum*, *L. excipuliforme*, and *Mycenastrum corium*, where classification approached perfection. Only Lycoperdon perlatum and *L. molle* exhibited marginally lower AUC values, suggesting slightly greater difficulty in differentiating these species. Taken together, these findings confirm the Swin Transformer’s capacity for high-precision classification, reliable discrimination, and broad generalizability in complex biological image analysis. The model consistently delivers stable, trustworthy predictions, even when challenged by visually similar categories.

### 3.5. MaxViT (Small TF 224)

The MaxViT architecture reached 84.16% accuracy on the test set, maintaining a reliable predictive profile across all classification scenarios. Precision and recall, both measured at 0.85, indicated dependable identification of positive cases while limiting false positives, as reflected by an *F*1-score of 0.84 (Figure 7a,b). Despite this steadiness, MaxViT struggled to resolve nuanced boundaries among morphologically similar species, with its discriminative capacity lagging behind that of more advanced neural paradigms. Although the model’s hybrid composition facilitated the recognition of diverse visual cues, its overall efficacy did not align with that of the top performers in multi-class biological image analysis. Taken together, these findings suggest that for datasets characterized by pronounced biological diversity and subtle visual distinctions, architectures specifically tailored for fine-grained discrimination may offer additional benefits.

As shown in the confusion matrix (Figure 7a), the MaxViT model maintained high classification accuracy for the majority of puffball species. Predictions for species such as *Apioperdon pyriforme*, *Lycoperdon echinatum*, and *Mycenastrum corium* were exact. At the same time, some cross-classification errors became apparent, particularly between visually similar types like *Lycoperdon molle* and *Lycoperdon perlatum*. Although overall error rates remained low, a tendency for increased confusion was observed among specific class pairs. The ROC analysis presented in the figure (Figure 7b) corroborates these findings; AUC values generally ranged from 0.97 to 1.00, confirming the model’s consistent ability to distinguish positive from negative cases effectively. Performance was strongest for species such as *Lycoperdon echinatum* and *Mycenastrum corium*. In contrast, marginally lower AUC scores for *Lycoperdon perlatum* and *Bovistella utriformis* indicated slightly weaker class separation for these particular taxa. Taken together, these results suggest that MaxViT delivers reliable classification performance and solid discriminative power in complex biological image analysis. However, its effectiveness diminishes somewhat when faced with classes exhibiting substantial morphological similarity, and it does not fully match the discrimination capabilities of other advanced convolutional and transformer models in these challenging cases.

## 4. Discussion

This study benchmarked five cutting-edge deep learning models to classify eight puffball species that are morphologically similar, combining convolutional and transformer-based architectures. The ConvNeXt-Base model emerged as the top performer, with an accuracy of 95.41%, a precision of 0.96, a recall of 0.95, and an *F*1-score of 0.95, outperforming all other models by a significant margin. The Swin Transformer came second, with an accuracy of 92.08% and balanced secondary metrics (precision and recall: 0.93). In contrast, the Vision Transformer (ViT Base Patch16) and MaxViT showed more moderate performances, with accuracy rates of 83.75% and 84.16%, respectively, each achieving an *F*1-score of 0.84. The EfficientNet-B3 model yielded the worst results across all metrics, with an accuracy of 82.08%, reflecting its relatively limited capacity to capture subtle inter-class distinctions. Comparing ConvNeXt-Base with the weakest model (EfficientNet-B3), a performance gap of 13.33% in accuracy and 13 percentage points in *F*1-score was evident. Additionally, ConvNeXt reduced misclassifications across all species, achieving near-perfect AUC values (0.99–1.00) and particularly excelling with visually ambiguous taxa such as *Mycenastrum corium* and *Lycoperdon excipuliforme*, highlighting its robust feature extraction ability in biologically complex datasets.

The deep learning architectures employed in this study exhibit distinct inductive biases, which contributed to their varying classification performances. ConvNeXt-Base, with its modern convolutional design, retained the local feature extraction strengths of traditional CNNs while benefiting from faster convergence, leading to higher accuracy under limited dataset conditions. In contrast, attention-based models such as Vision Transformer (ViT) and Swin Transformer, which typically require larger datasets and longer training schedules to fully leverage their global attention mechanisms, showed reduced performance in this fine-grained fungal classification task. Although EfficientNet-B3 is highly parameter-efficient, its relatively compact structure may have limited its ability to capture subtle morphological differences between visually similar species. These findings clearly demonstrate the impact of architectural design on classification success in domain-specific image analysis tasks.

The comparative evaluation of the models revealed distinct trends based on architectural design. Traditional CNNs, such as ConvNeXt-Base, excelled at extracting spatial hierarchies crucial for distinguishing fine morphological traits [17]. In contrast, transformer-based architectures like ViT and MaxViT demonstrated diminished performance, likely due to their greater dependence on large datasets and inherently weaker inductive biases [17,18]. The Swin Transformer, with its hybrid structure incorporating localized attention and hierarchical processing, closely rivalled ConvNeXt in accuracy, suggesting its enhanced adaptability to multi-scale biological patterns [19,20]. Despite their novelty, neither ViT nor MaxViT surpassed the convolutional models, indicating that for small to mid-sized datasets with closely related visual classes, convolutional backbones may still offer superior feature discrimination [21,22,23]. Furthermore, EfficientNet’s relatively shallow depth and dimension-scaling strategy, while computationally efficient, appeared inadequate for this high-granularity classification problem [24,25]. Uniform preprocessing, augmentation, and training conditions ensured that these performance differences stemmed from the intrinsic capabilities of the models rather than experimental variance [26,27].

Beyond this study, deep learning has increasingly been applied to fungal taxonomy and image-based species recognition [8,9,28]. Prior works have implemented CNNs or transfer learning methods to identify macrofungi, often focusing on broader genus-level classification or the distinction between edible and toxic species [28,29]. However, few studies have addressed intra-genus differentiation among gasteroid taxa, where morphological overlaps are common and expert-level interpretation has traditionally been required. The current hybrid CNN–SOM and transformer-inclusive strategy has demonstrated superior accuracy and generalization, particularly with rare or misclassified species [30,31,32]. Moreover, the inclusion of Mycenastrum corium [33], a taxonomically contentious species, underscores the model’s capacity to manage phylogenetic outliers. Comparable works employing explainable AI techniques, such as Grad-CAM or Score-CAM, are increasingly used to visualize decision boundaries and interpret fungal classification [34,35]. This aligns with our methodology and reinforces the growing integration of deep learning in taxonomic workflows [36,37,38]. In parallel, the core architectures utilized in this study Vision Transformer (ViT), EfficientNet-B3, and ConvNeXt-Base have also demonstrated utility in broader machine learning domains. ViT models have been successfully applied across various areas, including medical imaging, document classification, and vision-language tasks, often in combination with models such as BART for automated report generation [39]. EfficientNet, known for its compound scaling strategy, has been widely employed in biomedical image analysis, including in the classification of neurodegenerative diseases such as Alzheimer’s [40]. ConvNeXt architectures, with their modernized convolutional design, have also shown strong performance in visual interpretation tasks such as automated medical reporting [41]. While these architectures offer demonstrated flexibility, our results suggest that their relative effectiveness in fungal classification is strongly influenced by dataset size, domain-specific structure, and inter-class visual similarity.

## 5. Conclusions

The present study offers a comprehensive evaluation of deep learning architectures for the image-based classification of eight ecologically and taxonomically significant puffball species. The proposed methodology seamlessly integrates CNN and transformer-based approaches, establishing a robust framework for addressing fine-grained morphological distinctions that often challenge traditional mycological taxonomy. Among the tested models, ConvNeXt-Base achieved the highest classification performance with 95.41% accuracy, highlighting its strength in feature extraction and class discrimination. The Swin Transformer also demonstrated impressive performance (92.08% accuracy), underscoring the advantages of hierarchical attention mechanisms in biological image recognition. Models such as ViT and MaxViT, while competitive, were slightly less effective in distinguishing closely related taxa, stressing the importance of appropriate architecture selection in high-similarity biological datasets. These findings affirm the effectiveness of deep learning, particularly convolutional models, in managing taxonomic complexity and enhancing fungal biodiversity assessments.

The successful application of deep learning to puffball identification opens pathways for broader applications in fungal ecology, taxonomy, and conservation. The framework could be expanded to include more species, additional ecological metadata (e.g., substrate, habitat), and larger image repositories. Future studies may benefit from integrating explainable AI components such as Grad-CAM or Score-CAM to support biological interpretation by revealing species-specific visual patterns used by the models. These capabilities may assist mycologists in uncovering novel diagnostic features, improving field identification tools, and reducing reliance on labor-intensive microscopy or molecular methods. Moreover, the potential for integrating this system into mobile applications and citizen science platforms promises to democratize fungal biodiversity monitoring, particularly in under-sampled regions.

Looking ahead, future research should address current limitations by expanding the dataset size, diversifying imaging conditions (e.g., lighting, angles, backgrounds), and incorporating 3D or microscopic data. Additionally, multi-modal models that combine visual, ecological, and molecular inputs may further enhance classification accuracy and reliability, particularly for cryptic or morphologically ambiguous taxa. Finally, exploring lightweight architectures and model compression techniques will be crucial for enabling real-time deployment in field-based and resource-limited environments. Overall, this study affirms the transformative potential of deep learning in fungal taxonomy and establishes a solid foundation for interdisciplinary innovation in computational mycology and biodiversity science.

## Figures and Tables

**Figure 1 biology-14-00816-f001:**
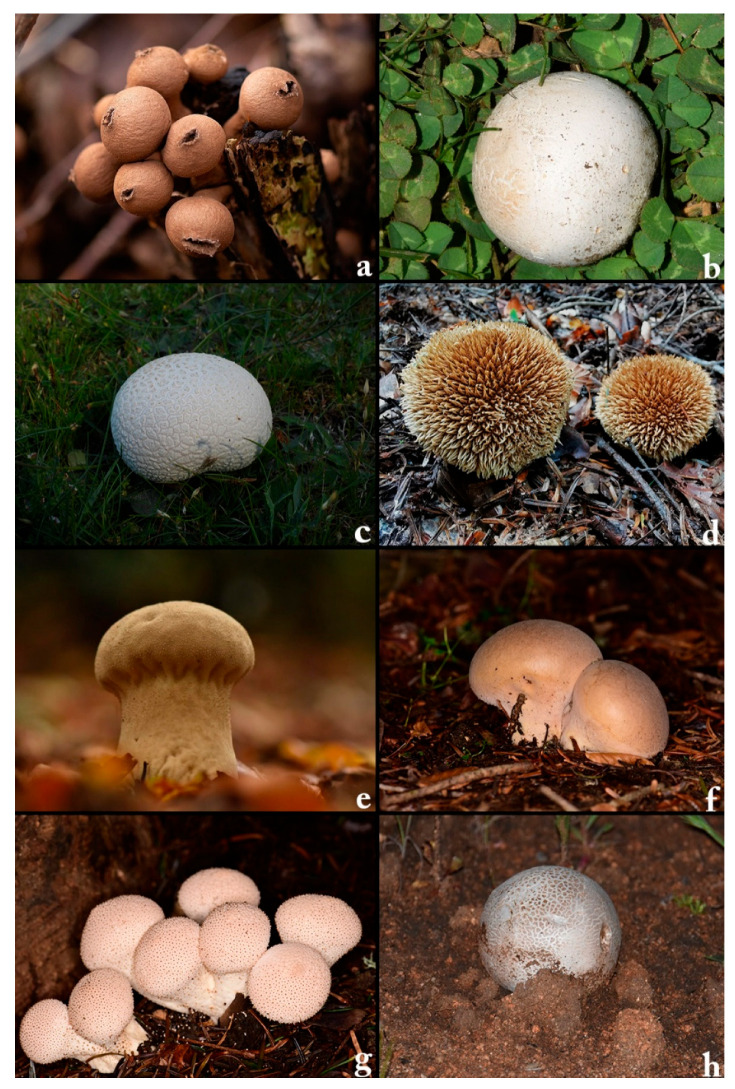
The dataset employed in this study includes eight distinct macrofungal species: (**a**) *Apioperdon pyriforme*, (**b**) Bovista plumbea, (**c**) *Bovistella utriformis*, (**d**) *Lycoperdon echinatum*, (**e**) *Lycoperdon excipuliforme*, (**f**) *Lycoperdon molle*, (**g**) *Lycoperdon perlatum*, and (**h**) *Mycenastrum corium*.

**Figure 2 biology-14-00816-f002:**
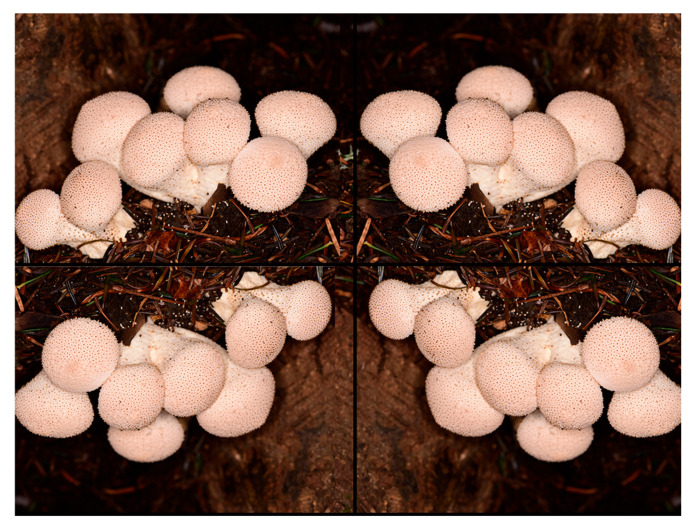
Sample data augmentation for puffball images.

**Figure 3 biology-14-00816-f003:**
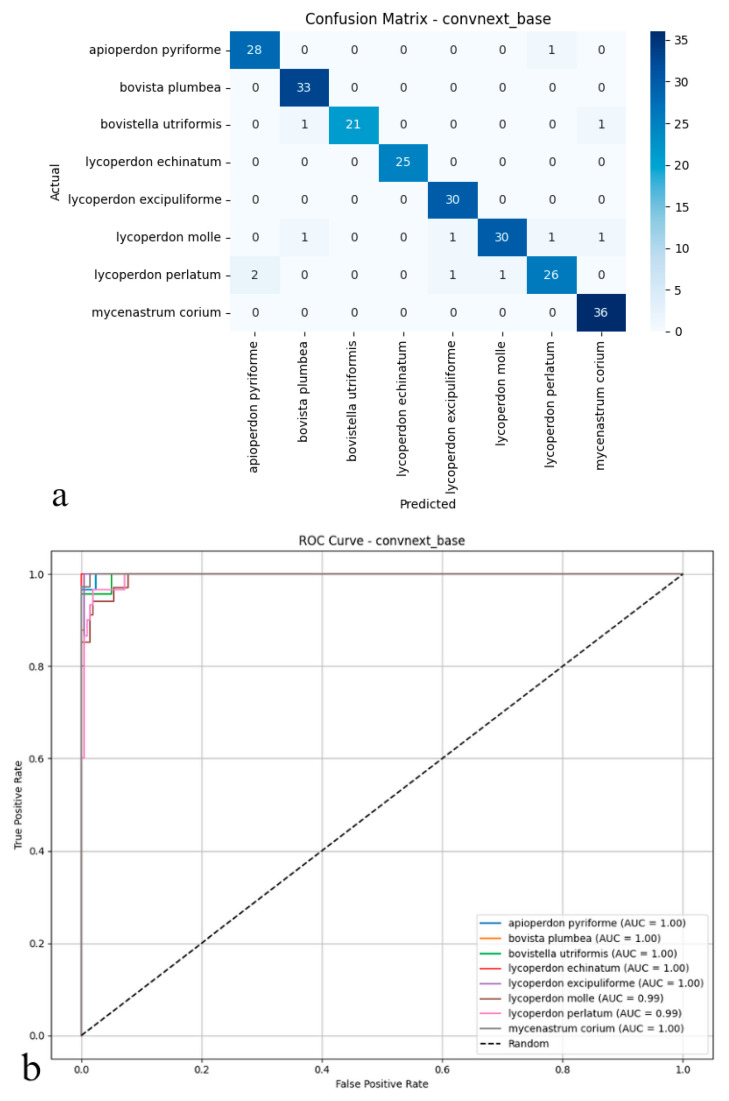
(**a**) Confusion matrix and (**b**) ROC curves with AUC values of the ConvNeXt-Base model for each puffball species.

**Figure 4 biology-14-00816-f004:**
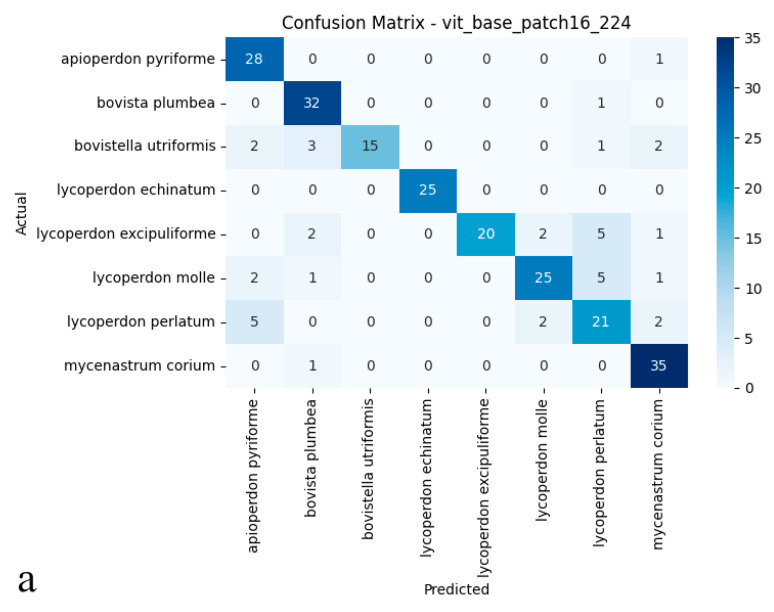
(**a**) Confusion matrix and (**b**) ROC curves with AUC values of the Vision Transformer (ViT Base Patch16) model for each macrofungus species.

**Figure 5 biology-14-00816-f005:**
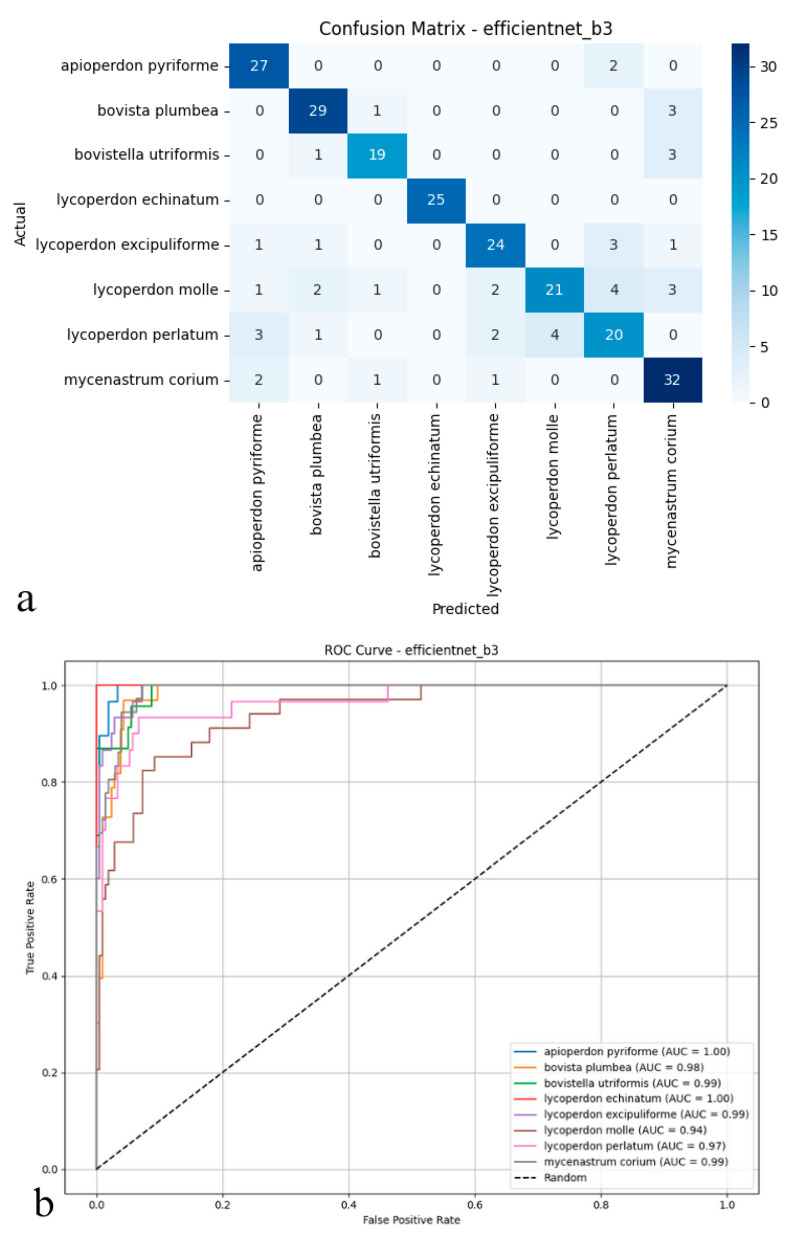
(**a**) Confusion matrix and (**b**) ROC curves with AUC values of the EfficientNet-B3 model for each macrofungus species.

**Figure 6 biology-14-00816-f006:**
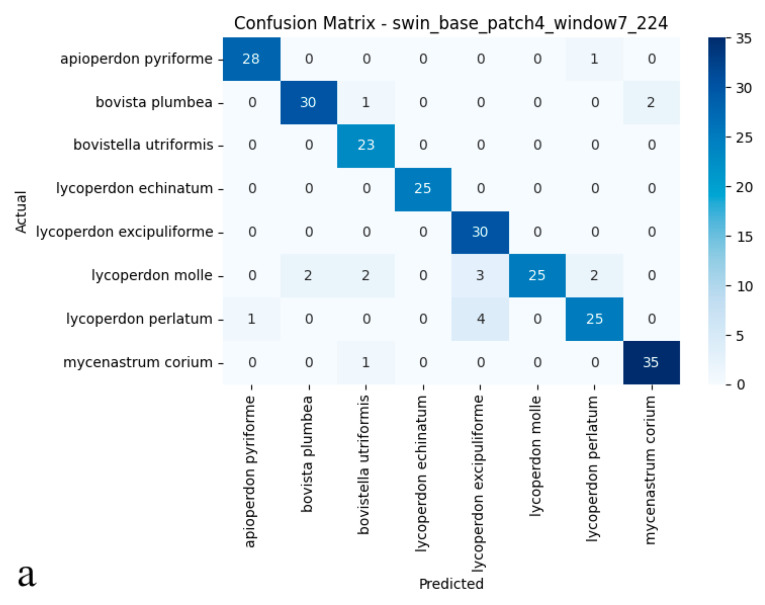
(**a**) Confusion matrix and (**b**) ROC curves with AUC values of the Swin Transformer (Small TF 224) model for each puffball species.

**Figure 7 biology-14-00816-f007:**
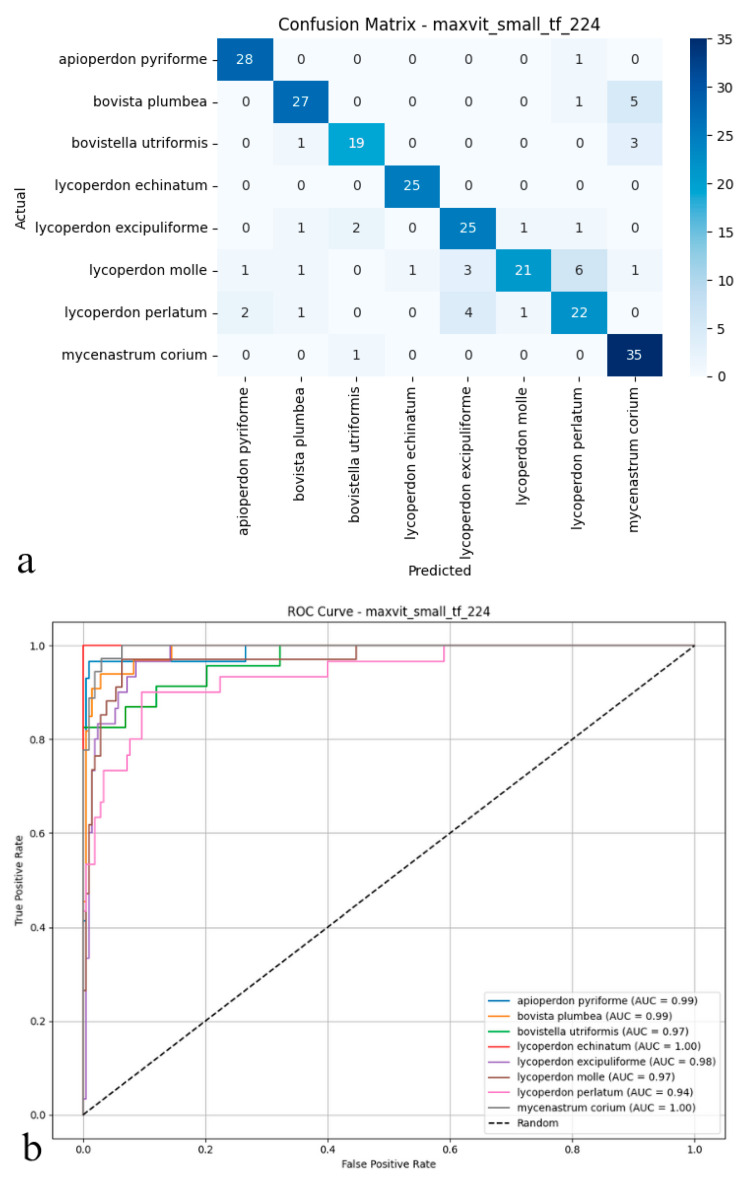
(**a**) Confusion matrix and (**b**) ROC curves with AUC values of the MaxViT (Small TF 224) model for each macrofungus species.

**Table 1 biology-14-00816-t001:** Comparative evaluation of five deep learning architectures; ConvNeXt-Base, Swin Transformer, Vision Transformer (ViT), MaxViT, and EfficientNet-B3 on the puffball species image dataset.

Base Model	Accuracy	Precision	Recall	*F*1-Score
**ConvNeXt-Base**	**0.95**	**0.96**	**0.95**	**0.95**
Vision Transformer (ViT Base Patch16)	0.84	0.86	0.83	0.84
EfficientNet-B3	0.82	0.83	0.83	0.82
**Swin Transformer (Small TF 224)**	**0.92**	**0.93**	**0.93**	**0.92**
MaxViT (Small TF 224)	0.84	0.85	0.85	0.84

The bold values indicate the highest performance achieved.

## Data Availability

Article data can be accessed from the responsible authors upon request, within a reasonable framework.

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
