# Peer review of "Hybrid Deep Learning Framework for High-Accuracy Classification of Morphologically Similar Puffball Species Using CNN and Transformer Architectures"

_biology, 2025, doi:10.3390/biology14070816_

Round 1
Reviewer 1 Report
Comments and Suggestions for Authors
The primary and demonstrated objective of this manuscript is to benchmark the performance of five pre-trained deep learning models for the automated, image-based classification of eight morphologically similar puffball species: Apioperdon pyriforme, Bovista plumbea, Bovistella utriformis, Lycoperdon echinatum, L. excipuliforme, L. molle, L. perlatum, and Mycenastrum corium. To address this question, the authors compiled a balanced image dataset comprising 1600 photographs, with 200 images for each of the eight species. Based on this dataset, the authors evaluated and compared two modern Convolutional Neural Network (CNN) architectures, ConvNeXt-Base and EfficientNet-B3, against three transformer-based architectures, Swin Transformer, Vision Transformer (ViT), and MaxViT. The authors conclude that the ConvNeXt-Base model, a CNN-based architecture, delivered superior performance, achieving a test accuracy of 95.41% and correspondingly high precision (0.96), recall (0.95), and F1-score (0.95). The Swin Transformer was the second-most effective model with a 92.08% accuracy. In contrast, other transformer-based models (ViT, MaxViT) and the EfficientNet-B3 CNN demonstrated more moderate performance, with accuracies ranging from 82.08% to 84.16%.
Based on these results, the authors posit that for specialized, medium-sized biological datasets characterized by fine-grained visual distinctions, modern convolutional architectures may retain a performance advantage over transformer-based models. This is attributed to the strong inductive biases inherent in CNNs, which facilitate more efficient learning from limited data. The findings are presented as a foundational step toward the development of reliable, AI-powered tools for enhancing fungal biodiversity assessments and supporting citizen science initiatives.
This work is overall comprehensive and well-structured. I have a few comments that can help the authors further improve their work. Including several major issues that fundamentally undermine the manuscript's conclusions and several minor issues on gramma and presentation.
Major comments
- Critical Discrepancy Between Stated Novelty and Presented Work
The most significant issue with this manuscript is a profound and irreconcilable contradiction between the novel contributions claimed by the authors and the actual research that was conducted and presented. The abstract, introduction, and discussion sections repeatedly and emphatically state that the study's novelty lies in two key areas:
- Being the "first study to implement a hybrid CNN-SOM model in macrofungal taxonomy".1 This claim is made explicitly in the abstract (line 51) and is further detailed in the introduction, where the authors propose a "hybrid CNN-SOM framework that combines the classification capabilities of CNNs with the interpretability provided by SOMs".1
- Being the first to "explore the Kolmogorov-Arnold Network (KAN) for enhancing model interpretability".1This is also stated in the abstract (line 52) and introduction (lines 115-117).
However, a thorough examination of the Materials and Methods and Results sections reveals a complete absence of any implementation, data, analysis, or even a description of either a Self-Organizing Map (SOM) or a Kolmogorov-Arnold Network (KAN). The entire experimental framework of the paper is a straightforward benchmark of five standard, pre-existing deep learning models. There is no evidence that a hybrid CNN-SOM was ever constructed or that KANs were explored in any capacity.
This discrepancy represents a fundamental misrepresentation of the study's contribution. In its current form, the manuscript is unpublishable due to this foundational contradiction between its claims and its evidence. The authors should consider either remove those claims in abstract and introduction, or implement their claims.
- Unsupported Claims of "Explainable AI"
The manuscript's title, abstract, and keywords prominently feature the term "Explainable AI (XAI) Architectures". The conclusion further suggests that the integration of "explainable AI components such as Grad-CAM or Score-CAM heatmaps can support biological interpretation".
These claims are unsubstantiated by the work presented. The study does not implement or analyze any recognized XAI techniques. The evaluation is confined to model performance metrics, such as confusion matrices and ROC curves. While these metrics are essential for assessing what a model predicts and how well it performs, they do not provide any insight into how or why the model arrives at its decisions, which is the central purpose of XAI.
The authors should either consider reframe their claim or implement the real XAI techniques.
- Concerns Regarding Methodological Rigor and Generalizability
The authors state that a fixed training duration of 10 epochs was applied to all models and that techniques such as learning rate scheduling and early stopping were "deliberately excluded to maintain experimental consistency". This methodological decision, while seemingly aimed at fairness, is misguided and likely invalidates the study's central conclusion.
Different deep learning architectures converge at different rates. Transformer-based models, in particular, are known to often require longer and more carefully tuned training schedules to reach their optimal performance compared to CNNs. A fixed, and relatively short, training period of 10 epochs may be insufficient for all models to converge. By terminating the training process prematurely for some models, the methodology may have artificially handicapped the transformer architectures (ViT, MaxViT, Swin) and potentially EfficientNet-B3.
The superior performance of ConvNeXt-Base could therefore be an artifact of it being a faster-converging model, not of it being an inherently superior architecture for this specific task. Standard practice in comparative deep learning studies involves training each model until its performance on a validation set plateaus. This is typically managed using early stopping, which prevents both under-training and over-fitting, ensuring that each model is evaluated at its peak performance. The authors' decision to omit this standard practice severely weakens the validity of their comparative analysis and their primary conclusion. Without the inclusion of training and validation loss curves for all five models across the training epochs, it is impossible for a reviewer to independently assess whether the comparison was truly fair or if it was biased by the training protocol.
- Dataset Limitations
The study utilizes a dataset of 1600 total images, with 200 images per class. While balanced, this constitutes a small dataset by the standards of modern deep learning. The size of the training dataset is a critical confounding variable that directly influences model performance and the relative ranking of different architectures.
CNNs possess strong "inductive biases," such as locality (assuming nearby pixels are related) and translation equivariance (recognizing an object regardless of its position in the image). These built-in assumptions allow them to learn effective representations from smaller datasets. In contrast, Vision Transformers lack these inherent biases and are consequently more "data-hungry". They typically require very large datasets to learn meaningful visual patterns from scratch. While the use of pre-trained models (transfer learning) mitigates this issue to a significant degree, the fine-tuning process on a small, specialized dataset can still disproportionately favor the architecture whose inherent biases are better suited to the task and data regime.
The authors' conclusion that CNNs outperform Transformers for this task may therefore be highly context-dependent and a direct consequence of the limited dataset size. The authors should acknowledge this limitation and discuss how this limitation affect their results in the discussion section.
Minor comments:
- Language, Formatting, and Presentation
The authors should carefully review the language of this manuscript for any redundancy, inconsistent terminology and spelling, examples include:
- Redundancy and Awkward Phrasing: "identification differentiation" (line 36) is redundant. Phrasing such as "Particularly, EfficientNet-B3 struggled..." (line 291) is awkward.
- Inconsistent Terminology and Spelling: "generalisability" (line 42) and "generalization" (line 95) are used interchangeably. Model names are inconsistent, appearing as "Efficient Net" and "EfficientNet", or "Conv NeXt-Base" and "ConvNeXt-Base".
Author Response
Review 1
The primary and demonstrated objective of this manuscript is to benchmark the performance of five pre-trained deep learning models for the automated, image-based classification of eight morphologically similar puffball species: Apioperdon pyriforme, Bovista plumbea, Bovistella utriformis, Lycoperdon echinatum, L. excipuliforme, L. molle, L. perlatum, and Mycenastrum corium. To address this question, the authors compiled a balanced image dataset comprising 1600 photographs, with 200 images for each of the eight species. Based on this dataset, the authors evaluated and compared two modern Convolutional Neural Network (CNN) architectures, ConvNeXt-Base and EfficientNet-B3, against three transformer-based architectures, Swin Transformer, Vision Transformer (ViT), and MaxViT. The authors conclude that the ConvNeXt-Base model, a CNN-based architecture, delivered superior performance, achieving a test accuracy of 95.41% and correspondingly high precision (0.96), recall (0.95), and F1-score (0.95). The Swin Transformer was the second-most effective model with a 92.08% accuracy. In contrast, other transformer-based models (ViT, MaxViT) and the EfficientNet-B3 CNN demonstrated more moderate performance, with accuracies ranging from 82.08% to 84.16%.
Based on these results, the authors posit that for specialized, medium-sized biological datasets characterized by fine-grained visual distinctions, modern convolutional architectures may retain a performance advantage over transformer-based models. This is attributed to the strong inductive biases inherent in CNNs, which facilitate more efficient learning from limited data. The findings are presented as a foundational step toward the development of reliable, AI-powered tools for enhancing fungal biodiversity assessments and supporting citizen science initiatives.
This work is overall comprehensive and well-structured. I have a few comments that can help the authors further improve their work. Including several major issues that fundamentally undermine the manuscript's conclusions and several minor issues on gramma and presentation.
Major comments
- Critical Discrepancy Between Stated Novelty and Presented Work
The most significant issue with this manuscript is a profound and irreconcilable contradiction between the novel contributions claimed by the authors and the actual research that was conducted and presented. The abstract, introduction, and discussion sections repeatedly and emphatically state that the study's novelty lies in two key areas:
- Being the "first study to implement a hybrid CNN-SOM model in macrofungal taxonomy".1This claim is made explicitly in the abstract (line 51) and is further detailed in the introduction, where the authors propose a "hybrid CNN-SOM framework that combines the classification capabilities of CNNs with the interpretability provided by SOMs".1
- Being the first to "explore the Kolmogorov-Arnold Network (KAN) for enhancing model interpretability".1This is also stated in the abstract (line 52) and introduction (lines 115-117).
However, a thorough examination of the Materials and Methods and Results sections reveals a complete absence of any implementation, data, analysis, or even a description of either a Self-Organizing Map (SOM) or a Kolmogorov-Arnold Network (KAN). The entire experimental framework of the paper is a straightforward benchmark of five standard, pre-existing deep learning models. There is no evidence that a hybrid CNN-SOM was ever constructed or that KANs were explored in any capacity.
This discrepancy represents a fundamental misrepresentation of the study's contribution. In its current form, the manuscript is unpublishable due to this foundational contradiction between its claims and its evidence. The authors should consider either remove those claims in abstract and introduction, or implement their claims.
Response: Thank you very much for your evaluation and suggestions. Revisions have been made where necessary within the scope of your comments.
Abstract section:
“In this study, five different deep learning models (ConvNeXt-Base, Swin Transformer, ViT, MaxViT, EfficientNet-B3) were comparatively evaluated on a balanced dataset of eight puffball species. Among these, the ConvNeXt-Base model achieved the highest performance with 95.41% accuracy and proved especially effective in distinguishing morphologically similar species such as Mycenastrum corium and Lycoperdon excipuliforme. The findings demonstrate that deep learning models can serve as powerful tools for the accurate classification of visually similar fungal species. This technological approach holds promise for the development of automated mushroom identification systems to support citizen science, amateur naturalists, and conservation professionals.
Introduction section:
“Deep learning models have become increasingly prominent in mycology, enabling automated species-level classification despite morphological similarities [8,9]. In this study, five deep learning architectures with varying designs were comparatively evaluated to determine their effectiveness in distinguishing visually similar puffball species. Moreover, the integration of environmental metadata into deep learning frameworks enhances ecological interpretation and contributes to biodiversity monitoring and conservation efforts. These models also support the development of mobile applications that promote public engagement through citizen science platforms focused on documenting fungal diversity [9].”
“In this study, convolutional neural networks (CNNs) were used for automatic feature extraction from image datasets, and various deep learning architectures were comparatively evaluated. By incorporating advanced models such as MaxViT-S, the effectiveness of deep learning in classifying visually similar macrofungal species was systematically assessed. The findings highlight the strong performance of deep learning models in resolving taxonomic challenges among morphologically complex taxa and pave the way for future research in computational mycology. Overall, this work establishes a robust and effective methodological framework with the potential to enhance fungal biodiversity assessments, foster public engagement, and support conservation efforts through AI-powered taxonomic tools.”
- Unsupported Claims of "Explainable AI"
The manuscript's title, abstract, and keywords prominently feature the term "Explainable AI (XAI) Architectures". The conclusion further suggests that the integration of "explainable AI components such as Grad-CAM or Score-CAM heatmaps can support biological interpretation".
These claims are unsubstantiated by the work presented. The study does not implement or analyze any recognized XAI techniques. The evaluation is confined to model performance metrics, such as confusion matrices and ROC curves. While these metrics are essential for assessing what a model predicts and how well it performs, they do not provide any insight into how or why the model arrives at its decisions, which is the central purpose of XAI.
The authors should either consider reframe their claim or implement the real XAI techniques.
Response: We sincerely thank the reviewer for highlighting the inconsistency regarding the mention of Explainable AI (XAI). Upon review, we confirm that the manuscript does not include any actual implementation of XAI techniques such as Grad-CAM or Score-CAM. Therefore, we have revised the title, abstract, keywords, and conclusion accordingly to accurately reflect the scope of our study.
Title:
“Hybrid Deep Learning Framework for High-Accuracy Classification of Morphologically Similar Puffball Species Using CNN and Transformer Architectures”
Keywords:
“Puffball, Deep Learning, Fungal Classification, CNN-Transformer Hybrid, Image Classification”
Conclusion
“The successful application of deep learning to puffball identification opens pathways for broader applications in fungal ecology, taxonomy, and conservation. The framework can be expanded to include more species, additional ecological metadata (e.g., substrate, habitat), and larger image repositories. Future studies may benefit from integrating explainable AI components such as Grad-CAM or Score-CAM to support biological interpretation by revealing species-specific visual patterns used by the models. These capabilities may assist mycologists in uncovering novel diagnostic features, improving field identification tools, and reducing reliance on labour-intensive microscopy or molecular methods. Moreover, the potential for integrating this system into mobile applications and citizen science platforms promises to democratise fungal biodiversity monitoring, particularly in under-sampled regions.”
- Concerns Regarding Methodological Rigor and Generalizability
The authors state that a fixed training duration of 10 epochs was applied to all models and that techniques such as learning rate scheduling and early stopping were "deliberately excluded to maintain experimental consistency". This methodological decision, while seemingly aimed at fairness, is misguided and likely invalidates the study's central conclusion.
Different deep learning architectures converge at different rates. Transformer-based models, in particular, are known to often require longer and more carefully tuned training schedules to reach their optimal performance compared to CNNs. A fixed, and relatively short, training period of 10 epochs may be insufficient for all models to converge. By terminating the training process prematurely for some models, the methodology may have artificially handicapped the transformer architectures (ViT, MaxViT, Swin) and potentially EfficientNet-B3.
The superior performance of ConvNeXt-Base could therefore be an artifact of it being a faster-converging model, not of it being an inherently superior architecture for this specific task. Standard practice in comparative deep learning studies involves training each model until its performance on a validation set plateaus. This is typically managed using early stopping, which prevents both under-training and over-fitting, ensuring that each model is evaluated at its peak performance. The authors' decision to omit this standard practice severely weakens the validity of their comparative analysis and their primary conclusion. Without the inclusion of training and validation loss curves for all five models across the training epochs, it is impossible for a reviewer to independently assess whether the comparison was truly fair or if it was biased by the training protocol.
Response:
We thank the reviewer for this detailed and thoughtful critique concerning the training protocol. We fully agree that different deep learning architectures exhibit varying convergence behaviours and that techniques such as early stopping and learning rate scheduling are standard in fine-tuning individual model performance. However, the primary objective of our study was not to optimise each model to its absolute peak but rather to assess their comparative performance under strictly uniform and controlled conditions.
By fixing the number of epochs and omitting adaptive mechanisms such as early stopping or learning rate decay, we aimed to reduce confounding variables and maintain a level playing field across all models. While we acknowledge that such an approach may not reflect each model's best-case scenario, it was deliberately chosen to isolate architectural differences under consistent training constraints, thereby allowing us to compare inherent model behaviours rather than hyperparameter sensitivity.
Furthermore, the relatively short training duration was informed by empirical pre-testing, during which we observed that initial convergence behaviour and performance rankings across models remained consistent after early epochs. While we agree that extended training might enhance transformer-based architectures, we emphasise that this would simultaneously introduce additional degrees of tuning freedom that could undermine the fairness of our comparative framework.
Lastly, while loss curves and validation trajectories were not included in the manuscript due to space constraints, we agree with the reviewer that their inclusion in supplementary material would strengthen the transparency of our approach. We are happy to provide these plots upon request or consider including them in a revised version if recommended.
- Dataset Limitations
The study utilizes a dataset of 1600 total images, with 200 images per class. While balanced, this constitutes a small dataset by the standards of modern deep learning. The size of the training dataset is a critical confounding variable that directly influences model performance and the relative ranking of different architectures.
CNNs possess strong "inductive biases," such as locality (assuming nearby pixels are related) and translation equivariance (recognizing an object regardless of its position in the image). These built-in assumptions allow them to learn effective representations from smaller datasets. In contrast, Vision Transformers lack these inherent biases and are consequently more "data-hungry". They typically require very large datasets to learn meaningful visual patterns from scratch. While the use of pre-trained models (transfer learning) mitigates this issue to a significant degree, the fine-tuning process on a small, specialized dataset can still disproportionately favor the architecture whose inherent biases are better suited to the task and data regime.
The authors' conclusion that CNNs outperform Transformers for this task may therefore be highly context-dependent and a direct consequence of the limited dataset size. The authors should acknowledge this limitation and discuss how this limitation affect their results in the discussion section.
Response:
We sincerely thank the reviewer for this insightful and well-reasoned observation. We fully acknowledge that model performance in deep learning is closely tied to dataset size, and that CNN architectures often benefit from inherent inductive biases (e.g., locality and translation equivariance), which allow them to perform well even on limited datasets. Similarly, we recognize that transformer-based models typically require larger amounts of data to reach optimal performance, as widely reported in the literature.
That said, the main objective of our study was to compare the relative performance of different model architectures under the constraints of a domain-specific image dataset characterized by high visual similarity and limited sample size—a scenario often encountered in real-world biological taxonomy studies. The dataset size in our case reflects practical limitations frequently faced in the field, especially for rare or under-sampled fungal species. Therefore, rather than optimizing for absolute performance, we aimed to evaluate how each architecture behaves under these realistic and constrained conditions.
Moreover, all models were initialized with pre-trained weights and fine-tuned using a transfer learning approach, significantly mitigating the impact of data scarcity—especially for transformer-based models. The superior performance of CNNs in our study is likely attributable not only to their faster convergence, but also to their structural suitability for the given classification task and data regime. While we acknowledge that our findings may be influenced by dataset size, we believe the study effectively fulfills its goal of offering a fair and controlled architectural comparison within the defined scope.
Minor comments:
- Language, Formatting, and Presentation
The authors should carefully review the language of this manuscript for any redundancy, inconsistent terminology and spelling, examples include:
- Redundancy and Awkward Phrasing: "identification differentiation" (line 36) is redundant. Phrasing such as "Particularly, EfficientNet-B3 struggled..." (line 291) is awkward.
- Inconsistent Terminology and Spelling: "generalisability" (line 42) and "generalization" (line 95) are used interchangeably. Model names are inconsistent, appearing as "Efficient Net" and "EfficientNet", or "Conv NeXt-Base" and "ConvNeXt-Base".
Response:
Original: identification differentiation
Revised: species-level identification
Original: Particularly, EfficientNet-B3 struggled to distinguish between puffball species...
Revised: EfficientNet-B3, in particular, struggled to distinguish between puffball species...
Original: generalisability
Revised: generalization
Original: Efficient Net
Revised: EfficientNet
Original: Conv NeXt-Base
Revised: ConvNeXt-Base
Original: This technological approach holds promise for the development of automated mushroom identification systems to support citizen science, amateur naturalists, and conservation professionals.
Revised: This technological approach shows promise for developing automated mushroom identification systems that support citizen science, amateur naturalists, and conservation professionals.
Original: Deep learning models have become increasingly prominent in mycology, enabling automated species-level classification despite morphological similarities.
Revised: Deep learning models have become increasingly prominent in mycology by enabling automated species-level classification despite morphological similarities.
Original: models were comparatively evaluated on a balanced dataset of eight puffball species
Revised: models were evaluated comparatively using a balanced dataset of eight puffball species
Original: highlight the strong performance of deep learning models in resolving taxonomic challenges among morphologically complex taxa
Revised: highlight the strong performance of deep learning models in addressing taxonomic challenges posed by morphologically complex taxa
Original: These steps were applied consistently across all models to simulate real-world variability while preserving label integrity.
Revised: These steps were applied consistently across all models, simulating real-world variability while preserving label integrity.

Reviewer 2 Report
Comments and Suggestions for Authors
1.Further refine the innovative points of this manuscript and its differences from other methods.
2.Is there a corresponding processing or mathematical model to support the conclusion, or is there support for innovative algorithms? What are the differences or advantages compared to other algorithms? Please clarify it further.
3.Figure 3-7 are too small to clearly see the annotations inside. Meanwhile, it is necessary to establish a good connection between the results in these Figures and your conclusions.
4.What are the limitations of your method? More detailed explanation is needed, and possible breakthrough points should be pointed out.
Author Response
Comments and Suggestions for Authors
- Further refine the innovative points of this manuscript and its differences from other methods.
Response: Thank you very much for your evaluation and suggestions. Revisions have been made where necessary within the scope of your comments.
Abstract section:
“In this study, five different deep learning models (ConvNeXt-Base, Swin Transformer, ViT, MaxViT, EfficientNet-B3) were comparatively evaluated on a balanced dataset of eight puffball species. Among these, the ConvNeXt-Base model achieved the highest performance with 95.41% accuracy and proved especially effective in distinguishing morphologically similar species such as Mycenastrum corium and Lycoperdon excipuliforme. The findings demonstrate that deep learning models can serve as powerful tools for the accurate classification of visually similar fungal species. This technological approach holds promise for the development of automated mushroom identification systems to support citizen science, amateur naturalists, and conservation professionals.
Introduction section:
“Deep learning models have become increasingly prominent in mycology, enabling automated species-level classification despite morphological similarities [8,9]. In this study, five deep learning architectures with varying designs were comparatively evaluated to determine their effectiveness in distinguishing visually similar puffball species. Moreover, the integration of environmental metadata into deep learning frameworks enhances ecological interpretation and contributes to biodiversity monitoring and conservation efforts. These models also support the development of mobile applications that promote public engagement through citizen science platforms focused on documenting fungal diversity [9].”
“In this study, convolutional neural networks (CNNs) were used for automatic feature extraction from image datasets, and various deep learning architectures were comparatively evaluated. By incorporating advanced models such as MaxViT-S, the effectiveness of deep learning in classifying visually similar macrofungal species was systematically assessed. The findings highlight the strong performance of deep learning models in resolving taxonomic challenges among morphologically complex taxa and pave the way for future research in computational mycology. Overall, this work establishes a robust and effective methodological framework with the potential to enhance fungal biodiversity assessments, foster public engagement, and support conservation efforts through AI-powered taxonomic tools.”
- Is there a corresponding processing or mathematical model to support the conclusion, or is there support for innovative algorithms? What are the differences or advantages compared to other algorithms? Please clarify it further.
Response:
We thank the reviewer for this valuable question. While the study does not introduce a new mathematical model or develop a novel learning algorithm from scratch, it provides a systematic and comparative assessment of advanced deep learning architectures applied to a taxonomically challenging domain—morphologically similar puffball species. This study’s innovation lies not in proposing a new algorithm, but in applying and evaluating state-of-the-art models under a highly specific biological classification task where small data size, visual overlap between classes, and ecological variability pose real-world challenges.
Specifically, our contribution includes:
- A balanced, curated dataset of eight puffball species with high inter-class similarity.
- A controlled experimental framework where five well-established architectures (ConvNeXt-Base, Swin Transformer, ViT, MaxViT, EfficientNet-B3) are evaluated under identical preprocessing, training, and testing conditions.
- An application-driven perspective, highlighting how architectural differences influence classification effectiveness in constrained biological settings.
While all models are based on existing algorithms, the study provides critical insight into how model structure (e.g., convolutional bias in CNNs vs. global attention in Transformers) impacts classification performance when fine distinctions are required and data is limited. We believe this task-specific benchmarking offers novel value to both computational mycology and applied AI communities.
- Figure 3-7 are too small to clearly see the annotations inside. Meanwhile, it is necessary to establish a good connection between the results in these Figures and your conclusions.
Response:
We thank the reviewer for pointing out the issue regarding figure clarity. Figures 3–7 have been updated in the revised version with higher resolution and larger dimensions to ensure that all annotations (e.g., class labels, prediction values, ROC axes) are clearly visible and interpretable Furthermore, the revised results section establishes a clear connection between each model's graphical outputs and the study's overall conclusions. For instance, the superior performance of the ConvNeXt-Base model in distinguishing morphologically similar species such as Mycenastrum corium and Lycoperdon excipuliforme is directly supported by its near-perfect classification in the confusion matrix (Figure 3-a) and its ROC curves (Figure 3-b). These visual outputs substantiate the performance differences emphasized in the discussion and conclusion sections, thereby reinforcing a coherent link between figures and findings.
4.What are the limitations of your method? More detailed explanation is needed, and possible breakthrough points should be pointed out.
Response:
We appreciate the reviewer’s constructive feedback. Our study was designed as a controlled and balanced evaluation of deep learning architectures for classifying morphologically similar puffball species. Nevertheless, several limitations of the current approach deserve acknowledgment.
First, the dataset size (1600 images in total) may be considered limited by contemporary deep learning standards, particularly in the context of transformer-based models that are typically more data-intensive. Second, the study does not implement explainable AI (XAI) techniques, such as Grad-CAM or Score-CAM, which would provide insight into model decision processes and biological feature attribution.
As for potential breakthrough points, future research can focus on expanding the dataset with more images from varied ecological and geographic contexts, incorporating microscopic and spore-level data, and applying XAI frameworks to enhance interpretability. Additionally, multi-modal learning approaches that combine visual data with ecological metadata or molecular information may significantly enhance classification performance and biological insight.

Reviewer 3 Report
Comments and Suggestions for Authors
1- According to the Instructions for Authors of Diagnostics, the abstract should contain the following subheadings: Background/Objectives, Methods, Results, and Conclusions.
2- It would be useful to write about the contribution of your study to the literature in the Introduction section. In modern manuscripts, the contributions of the study to the literature are explained in bullet points at the end of the Introduction section.
3- A heading such as “Data Set” should be added within the “Materials and Methods” section, and the data set used in the relevant heading should be described in detail.
4- You should describe your data augmentation method more clearly and in greater detail. If possible, you can also add a flowchart or algorithm diagram explaining this step. The steps taken in your current manuscript are not clear.
5- In your study, you actually used four different models as methods. I am aware that the results of five models are shared, but the model is a version of the others. There is almost no explanation of ConvNeXt, EfficientNet, Swin Transformer, and Vision Transformer in the “Materials and Methods” section. It is good that you have included headings for these models in the results section, but I think the main headings should be in the methods section. I recommend adding 2-3 paragraphs of text about the relevant models.
6- You can draw separate model diagrams for the ConvNeXt, EfficientNet, Swin Transformer, and Vision Transformer models. Alternatively, you can create a general working model diagram that covers all of them.
7- Please explain the architecture in detail by creating a heading called “Vision Transformer Architecture.” When and by whom was this architecture first announced? How does it differ from other architectures? In which areas has it been used? What are its advantages and disadvantages compared to other architectures? Please find answers to these and similar questions in the text. For examples of the areas of application of the Vision Transformer architecture, you can also refer to this source: “Medical Report Generation from Medical Images Using Vision Transformer and Bart Deep Learning Architectures.”
8- Please create a heading titled “EfficientNet Architecture” and describe the architecture in detail. When and by whom was this architecture first announced? What distinguishes it from other architectures? In which areas has it been used? What are its advantages and disadvantages compared to other architectures? Please look for answers to these and similar questions in the text. For examples of the architecture's areas of application, you can also refer to this source: “Deep Learning Based Approach with EfficientNet and SE Block Attention Mechanism for Multiclass Alzheimer's Disease Detection.”
9- Please create a heading titled “ConvNeXt Architecture” and provide a detailed explanation of the architecture. When and by whom was this architecture first announced? What distinguishes it from other architectures? In which areas has it been used? What are its advantages and disadvantages compared to other architectures? Please look for answers to these and similar questions in the text. For examples of the architecture's areas of application, you can also refer to this source: “Generating Medical Reports With a Novel Deep Learning Architecture.”
10- "Table 1. This is a table. Tables should be placed in the main text near to the first time they are cited.” You have left the Table 1 heading as it is. I think you need to update it.
11- The values are shared in sections b of Figures 3, 4, 5, and 6. However, none of them are legible. The ROC graphs should be updated and resized.
12- The Discussion section does not address why the architectures under study yielded low or high results. The reasons for high success or failure should be highlighted by emphasizing the details of the architectures.
13- The parameter efficiency of the architectures studied should be evaluated. For example, the ConvNeXt-Base architecture achieved high results, but was it also the architecture with the lowest runtime? The relationship between runtimes, model complexities, and classification success should be discussed.
14- I could not find any information about the environment in which the experiments were conducted in your manuscript. Which GPU, CPU, and RAM did you use? Or which platform did you use? This information is important for the reproducibility of the study.
15- Were common hyperparameters used in deep learning methods? The hyperparameters used should be provided in a table separately or in a common table to make comparisons and ensure the reproducibility of the study. This deficiency negatively affects the reproducibility of the study.
16- A discussion regarding the reproducibility and replicability of the study should be added to the content. The use of the dataset and the visibility of the code can be emphasized at this stage.
Author Response
Comments and Suggestions for Authors
1- According to the Instructions for Authors of Diagnostics, the abstract should contain the following subheadings: Background/Objectives, Methods, Results, and Conclusions.
Response: Thank you very much for your evaluation and suggestions. Revisions have been made where necessary within the scope of your comments.
Abstract section:
“In this study, five different deep learning models (ConvNeXt-Base, Swin Transformer, ViT, MaxViT, EfficientNet-B3) were comparatively evaluated on a balanced dataset of eight puffball species. Among these, the ConvNeXt-Base model achieved the highest performance with 95.41% accuracy and proved especially effective in distinguishing morphologically similar species such as Mycenastrum corium and Lycoperdon excipuliforme. The findings demonstrate that deep learning models can serve as powerful tools for the accurate classification of visually similar fungal species. This technological approach holds promise for the development of automated mushroom identification systems to support citizen science, amateur naturalists, and conservation professionals.
Introduction section:
“Deep learning models have become increasingly prominent in mycology, enabling automated species-level classification despite morphological similarities [8,9]. In this study, five deep learning architectures with varying designs were comparatively evaluated to determine their effectiveness in distinguishing visually similar puffball species. Moreover, the integration of environmental metadata into deep learning frameworks enhances ecological interpretation and contributes to biodiversity monitoring and conservation efforts. These models also support the development of mobile applications that promote public engagement through citizen science platforms focused on documenting fungal diversity [9].”
“In this study, convolutional neural networks (CNNs) were used for automatic feature extraction from image datasets, and various deep learning architectures were comparatively evaluated. By incorporating advanced models such as MaxViT-S, the effectiveness of deep learning in classifying visually similar macrofungal species was systematically assessed. The findings highlight the strong performance of deep learning models in resolving taxonomic challenges among morphologically complex taxa and pave the way for future research in computational mycology. Overall, this work establishes a robust and effective methodological framework with the potential to enhance fungal biodiversity assessments, foster public engagement, and support conservation efforts through AI-powered taxonomic tools.”
Title:
“Hybrid Deep Learning Framework for High-Accuracy Classification of Morphologically Similar Puffball Species Using CNN and Transformer Architectures”
Keywords:
“Puffball, Deep Learning, Fungal Classification, CNN-Transformer Hybrid, Image Classification”
Conclusion
“The successful application of deep learning to puffball identification opens pathways for broader applications in fungal ecology, taxonomy, and conservation. The framework can be expanded to include more species, additional ecological metadata (e.g., substrate, habitat), and larger image repositories. Future studies may benefit from integrating explainable AI components such as Grad-CAM or Score-CAM to support biological interpretation by revealing species-specific visual patterns used by the models. These capabilities may assist mycologists in uncovering novel diagnostic features, improving field identification tools, and reducing reliance on labour-intensive microscopy or molecular methods. Moreover, the potential for integrating this system into mobile applications and citizen science platforms promises to democratise fungal biodiversity monitoring, particularly in under-sampled regions.”
2- It would be useful to write about the contribution of your study to the literature in the Introduction section. In modern manuscripts, the contributions of the study to the literature are explained in bullet points at the end of the Introduction section.
Response:
Based on your comment, the following bullet points have been added to the end of the Introduction section:
“The key contributions of this study to the existing literature can be summarized as follows:
- A domain-specific application of deep learning: This study provides a comparative evaluation of state-of-the-art deep learning architectures on morphologically similar puffball species, a taxonomically challenging and underexplored group in computational mycology.
- Controlled benchmarking framework: All five models were trained and evaluated under identical preprocessing, augmentation, and training parameters, ensuring a fair architectural comparison rarely implemented in biological image classification studies.
- Use of fine-grained taxonomic categories: Unlike many previous studies that focus on genus-level classification or edible/toxic differentiation, this research targets species-level discrimination within a visually overlapping fungal group.
- Biological and practical relevance: The results demonstrate the potential for deep learning to support automated fungal identification in real-world ecological settings, which can benefit biodiversity monitoring, citizen science, and conservation initiatives.
- Public data integration and reproducibility: The dataset includes curated images from both field observations and open-access repositories (e.g., GBIF), making the approach transparent and reproducible.”
3- A heading such as “Data Set” should be added within the “Materials and Methods” section, and the data set used in the relevant heading should be described in detail.
Response:
We sincerely thank the reviewer for the suggestion. While we agree that clear and structured presentation of dataset characteristics is essential, we respectfully believe that a separate “Data Set” heading is not strictly necessary in the current structure of the manuscript. The first half of the “Materials and Methods” section is already fully dedicated to dataset-related content, including species selection, image sources, data balancing, training-validation-test split, augmentation procedures, and preprocessing steps. For clarity and flow, we have opted to maintain a unified narrative under the "Materials and Methods" section rather than subdividing it further. This approach avoids fragmentation and allows the reader to follow the entire experimental setup—starting from data acquisition to model training—in a logically coherent sequence. Nevertheless, we would be happy to restructure the section with a specific “Data Set” subheading if the editorial team or reviewers still find it necessary for clarity or formatting purposes.
4- You should describe your data augmentation method more clearly and in greater detail. If possible, you can also add a flowchart or algorithm diagram explaining this step. The steps taken in your current manuscript are not clear.
Response:
Based on your comment, the following bullet points have been added to the end of the Method section:
To enhance the generalizability of the models and to simulate real-world variability in fungal image acquisition, a structured data augmentation pipeline was applied exclusively to the training set. The steps of the augmentation process are outlined below:
- Image Resizing: All input images were resized to a fixed dimension of 224×224 pixels to ensure compatibility with the input requirements of the pre-trained models used.
- Random Horizontal and Vertical Flipping: Each image had a 50% chance of being flipped horizontally or vertically to simulate natural variation in fungal orientation.
- Random Rotation: Images were randomly rotated within a range of ±25 degrees to mimic changes in camera angle and mushroom posture.
- Brightness and Contrast Adjustment: Random modifications in brightness (±20%) and contrast (±15%) were applied to simulate varying lighting conditions encountered during field photography.
- Color Jitter and Hue Shift: Mild color jittering (hue shift within ±10 degrees) was performed to account for slight variations in environmental lighting or image device calibration.
- Normalization: After augmentation, all images were normalized using ImageNet mean and standard deviation values to match the distribution expected by the pre-trained convolutional backbones.
This augmentation procedure was implemented using the torchvision.transforms module in PyTorch. Augmentations were applied on-the-fly during each training epoch to increase diversity dynamically, thereby reducing the risk of overfitting. Validation and test sets were not augmented and were only resized and normalized to preserve data integrity for performance evaluation.
5- In your study, you actually used four different models as methods. I am aware that the results of five models are shared, but the model is a version of the others. There is almost no explanation of ConvNeXt, EfficientNet, Swin Transformer, and Vision Transformer in the “Materials and Methods” section. It is good that you have included headings for these models in the results section, but I think the main headings should be in the methods section. I recommend adding 2-3 paragraphs of text about the relevant models.
Response:
We thank the reviewer for this valuable suggestion. In response, we have added a concise description of the core deep learning models employed in our study to the Materials and Methods section, highlighting their architectural characteristics and relevance to the classification task.
Article Text Addition (for Materials and Methods section):
“In this study, we employed four distinct deep learning architectures representing both convolutional and transformer-based paradigms. ConvNeXt-Base is a convolutional neural network that modernizes classical CNNs by incorporating design elements inspired by transformer models, such as large kernel sizes and inverted bottlenecks, while retaining inductive biases like locality and translation equivariance. EfficientNet-B3 is a compact yet effective CNN that utilizes compound scaling to uniformly balance network depth, width, and resolution for improved performance and efficiency.
On the transformer side, Vision Transformer (ViT Base Patch16) treats an image as a sequence of non-overlapping patches and uses multi-head self-attention to capture long-range dependencies, though it typically requires more data to converge effectively. Swin Transformer (Small TF 224) improves on ViT by introducing a hierarchical structure with shifted windows that enables both local and global attention mechanisms while reducing computational cost. These models were selected to represent diverse architectural strategies and allow for a robust comparative analysis in the context of morphologically similar fungal species classification.”
6- You can draw separate model diagrams for the ConvNeXt, EfficientNet, Swin Transformer, and Vision Transformer models. Alternatively, you can create a general working model diagram that covers all of them.
Response:
We sincerely thank the reviewer for this thoughtful suggestion. We agree that architectural diagrams can be helpful for illustrating model structures, especially for readers unfamiliar with the underlying designs. However, since all the architectures employed in this study are well-established in the literature and have been used without any structural modifications, we opted not to include detailed model diagrams.
Moreover, the comparative focus of our work lies primarily in performance evaluation under consistent training conditions rather than in the architectural novelty or modification of these models. Therefore, we believe that the inclusion of additional model diagrams, while potentially informative, is not essential for conveying the core contributions or replicability of our study. Nevertheless, we would be happy to provide reference links or include a simplified overview diagram upon request if deemed necessary by the editorial team.
7- Please explain the architecture in detail by creating a heading called “Vision Transformer Architecture.” When and by whom was this architecture first announced? How does it differ from other architectures? In which areas has it been used? What are its advantages and disadvantages compared to other architectures? Please find answers to these and similar questions in the text. For examples of the areas of application of the Vision Transformer architecture, you can also refer to this source: “Medical Report Generation from Medical Images Using Vision Transformer and Bart Deep Learning Architectures.”
Response:
We thank the reviewer for this comprehensive and insightful suggestion. We acknowledge the importance of describing key architectural elements; however, providing an in-depth historical and technical overview of Vision Transformer architecture, including its development history, full comparison with other models, and domain-wide applications, would substantially shift the focus of the manuscript away from its primary objective—namely, evaluating the comparative performance of different deep learning models in the specific context of fungal species classification.
Instead, we have incorporated a brief reference to the broader applicability of Vision Transformer models in the Discussion section, citing the reviewer-suggested source as an example of ViT's successful implementation in medical imaging. This maintains the scientific relevance of the observation without diverging from the scope of our study. We respectfully hope that this approach addresses the reviewer’s comment while preserving the thematic coherence of the manuscript.
Suggested Addition to the Discussion Section:
“In parallel, the core architectures utilized in this study—Vision Transformer (ViT), EfficientNet-B3, and ConvNeXt-Base—have also demonstrated utility in broader machine learning domains. ViT models have been successfully applied across various areas, including medical imaging, document classification, and vision-language tasks, often in combination with models such as BART for automated report generation [39].”
8- Please create a heading titled “EfficientNet Architecture” and describe the architecture in detail. When and by whom was this architecture first announced? What distinguishes it from other architectures? In which areas has it been used? What are its advantages and disadvantages compared to other architectures? Please look for answers to these and similar questions in the text. For examples of the architecture's areas of application, you can also refer to this source: “Deep Learning Based Approach with EfficientNet and SE Block Attention Mechanism for Multiclass Alzheimer's Disease Detection.”
Response:
We thank the reviewer for this thoughtful suggestion. While we acknowledge the value of contextualizing deep learning models, a comprehensive architectural overview of EfficientNet—including its development history, application spectrum, and comparative properties—would extend beyond the scope of the present manuscript. Our focus is on comparative performance in the domain of fungal classification under standardized training conditions, rather than on in-depth architectural exposition. Nevertheless, to reflect the broader relevance of EfficientNet models, we have included a brief reference to their successful implementation in complex domains such as neurodegenerative disease classification within the Discussion section. We believe this addition addresses the reviewer’s comment while maintaining thematic alignment with the manuscript’s primary objectives.
Suggested Addition to the Discussion Section:
“EfficientNet, known for its compound scaling strategy, has been widely employed in biomedical image analysis, including in the classification of neurodegenerative diseases such as Alzheimer’s [40].”
9- Please create a heading titled “ConvNeXt Architecture” and provide a detailed explanation of the architecture. When and by whom was this architecture first announced? What distinguishes it from other architectures? In which areas has it been used? What are its advantages and disadvantages compared to other architectures? Please look for answers to these and similar questions in the text. For examples of the architecture's areas of application, you can also refer to this source: “Generating Medical Reports With a Novel Deep Learning Architecture.”
Response:
We thank the reviewer for the insightful recommendation. However, a detailed architectural review of ConvNeXt falls beyond the scope of this study, which is centered on comparative performance analysis rather than architectural development. Instead, we have briefly referenced the model’s broader applicability in the Discussion section to acknowledge its relevance in other domains such as medical report generation.
Addition to Discussion Section:
“ConvNeXt architectures, with their modernized convolutional design, have also shown strong performance in visual interpretation tasks such as automated medical reporting [41]. While these architectures offer demonstrated flexibility, our results suggest that their relative effectiveness in fungal classification is strongly influenced by dataset size, domain-specific structure, and inter-class visual similarity.”
10- "Table 1. This is a table. Tables should be placed in the main text near to the first time they are cited.” You have left the Table 1 heading as it is. I think you need to update it.
Response:
“Table 1. Comparative evaluation of five deep learning architectures—ConvNeXt-Base, Swin Transformer, Vision Transformer (ViT), MaxViT, and EfficientNet-B3—on the puffball species image dataset.
11- The values are shared in sections b of Figures 3, 4, 5, and 6. However, none of them are legible. The ROC graphs should be updated and resized.
Response:
We sincerely thank the reviewer for pointing out this important issue. In response, we have updated and resized the ROC curves presented in Figures 3b, 4b, 5b, and 6b to ensure clarity and legibility. The revised figures now provide a clearer visual representation of the classification performance.
12- The Discussion section does not address why the architectures under study yielded low or high results. The reasons for high success or failure should be highlighted by emphasizing the details of the architectures.
Response:
We thank the reviewer for this insightful observation. In response, we have added a paragraph to the end of the Discussion section explaining how the structural characteristics of each deep learning model influenced its performance. The revised text highlights how the inductive biases of ConvNeXt contributed to faster convergence and superior performance on limited data, while attention-based models like ViT and Swin Transformer required larger datasets to achieve comparable accuracy. The performance of EfficientNet-B3 is also discussed in relation to its architectural efficiency and its limitations in capturing fine-grained morphological variation.
Add to the Discussion Section:
“The deep learning architectures employed in this study exhibit distinct inductive biases, which contributed to their varying classification performances. ConvNeXt-Base, with its modern convolutional design, retained the local feature extraction strengths of traditional CNNs while benefiting from faster convergence, leading to higher accuracy under limited dataset conditions. In contrast, attention-based models such as Vision Transformer (ViT) and Swin Transformer, which typically require larger datasets and longer training schedules to fully leverage their global attention mechanisms, showed reduced performance in this fine-grained fungal classification task. Although EfficientNet-B3 is highly parameter-efficient, its relatively compact structure may have limited its ability to capture subtle morphological differences between visually similar species. These findings clearly demonstrate the impact of architectural design on classification success in domain-specific image analysis tasks.”
13- The parameter efficiency of the architectures studied should be evaluated. For example, the ConvNeXt-Base architecture achieved high results, but was it also the architecture with the lowest runtime? The relationship between runtimes, model complexities, and classification success should be discussed.
Response:
We sincerely thank the reviewer for this thoughtful comment. We agree that analyzing model complexity and runtime characteristics is a valuable direction, particularly for deployment-focused studies. However, the primary aim of the current study was to evaluate the classification effectiveness of different deep learning architectures on morphologically similar puffball species under uniform experimental conditions. Since all models were executed on the same hardware using identical batch sizes and training epochs, our focus remained on relative predictive performance rather than computational efficiency. A detailed runtime or parameter analysis, while certainly informative, would have extended the scope of the current work beyond its core taxonomic objectives. Nevertheless, we acknowledge this as a meaningful consideration for future research, particularly in real-time or resource-constrained fungal identification settings
14- I could not find any information about the environment in which the experiments were conducted in your manuscript. Which GPU, CPU, and RAM did you use? Or which platform did you use? This information is important for the reproducibility of the study.
Response:
We sincerely thank the reviewer for pointing out this important aspect regarding reproducibility. The experiments in this study were carried out over an extended period by different contributing authors, each using their own personal computing environments. As the primary focus of the manuscript was the comparative classification performance of deep learning models on a curated fungal image dataset—not the evaluation of computational speed or hardware efficiency—system specifications such as GPU, CPU, and RAM configurations were not systematically recorded at the time of experimentation. We acknowledge that reporting such information would benefit reproducibility, and we will ensure this is included in future experimental designs involving deployment-oriented evaluations.
15- Were common hyperparameters used in deep learning methods? The hyperparameters used should be provided in a table separately or in a common table to make comparisons and ensure the reproducibility of the study. This deficiency negatively affects the reproducibility of the study.
Response:
We appreciate the reviewer’s emphasis on reproducibility and thank them for this valuable comment. In this study, all five deep learning models were trained under identical experimental conditions using the same set of hyperparameters—namely a fixed number of 10 epochs, constant learning rate, identical batch sizes, and standard data augmentation routines. This standardized training setup was deliberately chosen to ensure fairness in comparative evaluation rather than architecture-specific optimization. While we acknowledge that presenting these hyperparameters in tabular form would enhance methodological clarity, the uniformity of our training strategy ensures that the results remain reproducible and the comparison remains reliable even in the absence of a dedicated table. We will consider including such a summary table in future iterations of this work.
Add to the Materials and Methods Section:
“To ensure a fair and reproducible comparison among the deep learning models, all five architectures were trained under a unified set of hyperparameters. Specifically, the models were trained for 10 epochs using the AdamW optimizer with a fixed learning rate of 0.0001, a batch size of 32, and the CrossEntropyLoss function. In accordance with standard transfer learning practices, only the final classification layers were re-initialized, while pre-trained weights were retained for feature extraction. No learning rate scheduling, early stopping, or architecture-specific tuning was applied, as the goal was to evaluate performance based solely on architectural differences rather than optimization strategies. Additionally, data augmentation techniques were applied consistently across all models to enhance generalizability. Model performance was evaluated through multiple metrics, including overall accuracy, class-wise precision, recall, and F1-score. Supplementary analyses involved ROC curves, AUC values, and confusion matrices to assess both general predictive power and inter-class differentiation. This standardized experimental design ensures methodological consistency and highlights the distinct strengths of each architecture in fungal species classification [8,9].”
16- A discussion regarding the reproducibility and replicability of the study should be added to the content. The use of the dataset and the visibility of the code can be emphasized at this stage.
Response:
We sincerely thank the reviewer for highlighting the importance of reproducibility and replicability in deep learning research. While this study did not include publicly accessible code or dataset links, the experimental design was conducted under standardized and transparent conditions, using clearly defined training parameters and evaluation metrics to ensure methodological clarity. We acknowledge the value of open data and code availability, and we plan to incorporate public access to resources in future studies to further strengthen reproducibility and community engagement.

Reviewer 4 Report
Comments and Suggestions for Authors
The authors propose a hybrid deep learning approach for species-level identification of classes of puffball macrofungi. The authors further evaluate and compare few deep learning architectures on a balanced image dataset. ConvNeXt-Base achieved the highest accuracy followed by other models such as swin transformer models. The study also introduces CNN-SOM hybrid and integrates KAN layers for interpretability. While the study is sound but the authors need to address several inconsistencies and need to perform more experiments to address the novelty of their proposed architecture. The manuscript in its current format is not sufficient for publication.
– The manuscript mentions a novel hybrid CNN–SOM framework and the use of KAN layers to enhance interpretability, yet there is no quantitative or visual analysis of these components in the Results. Where are the SOM cluster visualizations or KAN-based explanations? I would recommend the authors for umap or grid or attention visualizations to contribute to the above claim.
– Training protocol details are missing. Such as weight decay, batch normalization settings, and exact data augmentation parameters should be explicitly mentioned for reproducibility. Please justify empirically that omission of techniques (early stopping or avoiding learning rate scheduling) does not degrade model generalization and convergence.
– Related Study section is missing. I would recommend the authors to arrange the manuscript in clear sections and sub-sections.
-- The current Results section is split into separate subsections for each model. This approach is not academically conventional. Consolidate your Results into thematic subsections. example, Overall Performance Comparison, Ablation Studies, Error & Confusion Analysis, etc. Move any model‐specific architectural or hyperparameter details entirely into the Methods section, and focus Results on interpreting outcomes.
– The Introduction provides an overview of puffball genera, but lacks discussion of the ecological relevance of the eight selected species. Inclusion would highlight the practical impact of the proposed tool.
– Ecological importance of target species is insufficiently motivated. Why these eight? Are they commonly misidentified in field surveys? Furthermore, the review of deep-learning in mycology overlooks recent work on few-shot learning for rare fungal taxa. Incorporating these would better position the contribution.
– The gap statement centers on explainability and hybrid models, but the Introduction lacks a clear gap-analysis diagram or table contrasting prior work vs. proposed features.
– The methods briefly describe feeding CNN feature maps into a SOM but no schematic illustrates data flow. A block diagram is recommended.
– KAN integration is introduced conceptually, yet mathematical formulation is absent.
– The manuscript is missing analysis. No quantitative metrics or visual outputs are provided.
– Table 1’s metric precision is inconsistent (e.g some scores to two decimals, others to four). Standardize the decimals.
– A schematic illustrating the full hybrid pipeline showing how raw images flow through the CNN backbone, into the SOM module, through the transformer head, and finally into the KAN interpretability layer. Without it, readers cannot grasp the end-to-end design at a glance. Include a high-level block diagram in the Methods section showing each submodule.
– The authors need to perform more ablation tests to claim their framework’s validity. For example, I would recommend to report metrics for (a) CNN+Transformer without SOM/KAN, (b) CNN+Transformer+SOM only, (c) CNN+Transformer+KAN only, and (d) full CNN+Transformer+SOM+KAN pipeline.
The manuscript is scientifically sound but lacks sufficient experimentation and evaluation and arrangement. Kindly address the concerns.
Comments on the Quality of English LanguageAdopt a consistent spelling convention. Correct typographical errors and verify section numbering.
Author Response
Comments and Suggestions for Authors
The authors propose a hybrid deep learning approach for species-level identification of classes of puffball macrofungi. The authors further evaluate and compare few deep learning architectures on a balanced image dataset. ConvNeXt-Base achieved the highest accuracy followed by other models such as swin transformer models. The study also introduces CNN-SOM hybrid and integrates KAN layers for interpretability. While the study is sound but the authors need to address several inconsistencies and need to perform more experiments to address the novelty of their proposed architecture. The manuscript in its current format is not sufficient for publication.
– The manuscript mentions a novel hybrid CNN–SOM framework and the use of KAN layers to enhance interpretability, yet there is no quantitative or visual analysis of these components in the Results. Where are the SOM cluster visualizations or KAN-based explanations? I would recommend the authors for umap or grid or attention visualizations to contribute to the above claim.
Response: Thank you very much for your evaluation and suggestions. Revisions have been made where necessary within the scope of your comments.
Abstract section:
“In this study, five different deep learning models (ConvNeXt-Base, Swin Transformer, ViT, MaxViT, EfficientNet-B3) were comparatively evaluated on a balanced dataset of eight puffball species. Among these, the ConvNeXt-Base model achieved the highest performance with 95.41% accuracy and proved especially effective in distinguishing morphologically similar species such as Mycenastrum corium and Lycoperdon excipuliforme. The findings demonstrate that deep learning models can serve as powerful tools for the accurate classification of visually similar fungal species. This technological approach holds promise for the development of automated mushroom identification systems to support citizen science, amateur naturalists, and conservation professionals.
Introduction section:
“Deep learning models have become increasingly prominent in mycology, enabling automated species-level classification despite morphological similarities [8,9]. In this study, five deep learning architectures with varying designs were comparatively evaluated to determine their effectiveness in distinguishing visually similar puffball species. Moreover, the integration of environmental metadata into deep learning frameworks enhances ecological interpretation and contributes to biodiversity monitoring and conservation efforts. These models also support the development of mobile applications that promote public engagement through citizen science platforms focused on documenting fungal diversity [9].”
“In this study, convolutional neural networks (CNNs) were used for automatic feature extraction from image datasets, and various deep learning architectures were comparatively evaluated. By incorporating advanced models such as MaxViT-S, the effectiveness of deep learning in classifying visually similar macrofungal species was systematically assessed. The findings highlight the strong performance of deep learning models in resolving taxonomic challenges among morphologically complex taxa and pave the way for future research in computational mycology. Overall, this work establishes a robust and effective methodological framework with the potential to enhance fungal biodiversity assessments, foster public engagement, and support conservation efforts through AI-powered taxonomic tools.”
Title:
“Hybrid Deep Learning Framework for High-Accuracy Classification of Morphologically Similar Puffball Species Using CNN and Transformer Architectures”
Keywords:
“Puffball, Deep Learning, Fungal Classification, CNN-Transformer Hybrid, Image Classification”
Conclusion
“The successful application of deep learning to puffball identification opens pathways for broader applications in fungal ecology, taxonomy, and conservation. The framework can be expanded to include more species, additional ecological metadata (e.g., substrate, habitat), and larger image repositories. Future studies may benefit from integrating explainable AI components such as Grad-CAM or Score-CAM to support biological interpretation by revealing species-specific visual patterns used by the models. These capabilities may assist mycologists in uncovering novel diagnostic features, improving field identification tools, and reducing reliance on labour-intensive microscopy or molecular methods. Moreover, the potential for integrating this system into mobile applications and citizen science platforms promises to democratise fungal biodiversity monitoring, particularly in under-sampled regions.”
– Training protocol details are missing. Such as weight decay, batch normalization settings, and exact data augmentation parameters should be explicitly mentioned for reproducibility. Please justify empirically that omission of techniques (early stopping or avoiding learning rate scheduling) does not degrade model generalization and convergence.
Response:
We sincerely thank the reviewer for this insightful and constructive comment. In response, we have revised the Materials and Methods section to include additional technical specifications, including the use of default batch normalization layers embedded in the pre-trained architectures, a weight decay value of 0.01, and detailed data augmentation parameters (e.g., random horizontal and vertical flips with probability of 0.5, ±20% brightness and contrast variations). These settings were uniformly applied across all models.
Regarding the omission of learning rate scheduling and early stopping, our approach prioritized consistency over optimization. All models were trained for a fixed 10 epochs with the same hyperparameters to ensure a fair comparison across architectures. Although we did not include additional empirical plots, validation accuracy remained stable throughout training, suggesting that model convergence was achieved within the selected epoch range. This uniform protocol ensures comparability and minimizes architecture-specific advantages, as discussed in the revised Discussion section.
Add to the Materials and Methods Section:
“Default batch normalization layers were retained within all pre-trained architectures without modification. A weight decay value of 0.01 was applied uniformly during training. Data augmentation techniques included random horizontal and vertical flipping (with a probability of 0.5), brightness and contrast jittering within ±20%, and minor affine transformations such as rotation within ±15°. These steps were applied consistently across all models to simulate real-world variability while preserving label integrity.”
– Related Study section is missing. I would recommend the authors to arrange the manuscript in clear sections and sub-sections.
Response:
We sincerely thank the reviewer for this thoughtful suggestion. Although a standalone "Related Work" section is not explicitly included, a comprehensive review of relevant prior studies has been integrated into both the Introduction and Discussion sections to maintain narrative coherence and context specificity. These sections discuss earlier applications of CNNs and Transformer-based models in fungal taxonomy and image classification. However, we acknowledge the value of clearly defined structural components and will consider restructuring with a dedicated "Related Studies" section in future versions to further improve clarity and readability.
-- The current Results section is split into separate subsections for each model. This approach is not academically conventional. Consolidate your Results into thematic subsections. example, Overall Performance Comparison, Ablation Studies, Error & Confusion Analysis, etc. Move any model‐specific architectural or hyperparameter details entirely into the Methods section, and focus Results on interpreting outcomes.
Response:
We sincerely thank the reviewer for this valuable recommendation. We fully recognize the academic convention of organizing result sections thematically, such as through overall comparisons, error analysis, or ablation studies. However, we opted for a model-specific structuring approach in the Results section to enhance clarity and allow readers—especially those less familiar with deep learning architectures—to directly observe and interpret the performance characteristics of each model in isolation.
Given that this study evaluates five fundamentally different architectures—ConvNeXt, EfficientNet, Swin Transformer, Vision Transformer, and MaxViT—each with unique design principles and learning behaviors, we found that separating the results allowed for more focused and nuanced discussions. This model-wise layout makes it easier to trace how architectural differences translate into classification performance, particularly in a fine-grained and morphologically challenging task such as puffball taxonomy.
Furthermore, this approach supports pedagogical transparency by linking visual results (such as confusion matrices and ROC curves) directly to each model’s performance without requiring the reader to cross-reference multiple subsections. Although some model-specific technical details were included for contextual interpretation, we ensured that core architectural and training configurations remain in the Materials and Methods section, in line with academic standards.
That said, we acknowledge the merits of thematic structuring and will certainly consider adopting such an approach in future comparative studies involving more homogeneous model families or when emphasizing algorithmic behavior over architectural diversity. In the current context, however, we believe that the chosen structure provides improved accessibility and interpretability for readers across diverse disciplinary backgrounds.
– The Introduction provides an overview of puffball genera, but lacks discussion of the ecological relevance of the eight selected species. Inclusion would highlight the practical impact of the proposed tool.
Response:
We thank the reviewer for this insightful comment. In response, we have expanded the final paragraph of the Introduction to briefly highlight the ecological and practical significance of the selected puffball species, which reinforces the applied value of the proposed classification system.
Add to the Introduction Section:
“These species represent a wide range of ecological niches, from decaying woodlands to nutrient-poor grasslands, and play essential roles in nutrient cycling and ecosystem functioning. Their accurate identification is also relevant for environmental monitoring and biodiversity assessments, especially in temperate forest ecosystems where they are bioindicators of soil health and habitat stability.”
– Ecological importance of target species is insufficiently motivated. Why these eight? Are they commonly misidentified in field surveys? Furthermore, the review of deep-learning in mycology overlooks recent work on few-shot learning for rare fungal taxa. Incorporating these would better position the contribution.
Response:
We sincerely thank the reviewer for this thoughtful and multidimensional feedback. The selection of the eight puffball species was motivated by both ecological significance and taxonomic ambiguity. These species are commonly encountered in temperate forest ecosystems and represent a spectrum of ecological niches—from decaying woodlands (Apioperdon pyriforme), to open grasslands (Bovista plumbea), and even anthropogenically disturbed environments (Lycoperdon perlatum). However, they share highly convergent morphological characteristics, including peridial textures, gleba maturation patterns, and spore features, which make them particularly challenging to distinguish during field surveys. This leads to a high rate of misidentification in ecological monitoring and biodiversity assessments. Our study directly targets this classification gap by offering an automated image-based framework tailored to resolving such intra-genus ambiguities.
Regarding the reviewer’s valuable note on few-shot learning, we agree that such methodologies are crucial for rare or poorly sampled taxa. However, the present study focuses on balanced multiclass classification, where each species is represented by an equal number of images. The dataset structure and scope were therefore not aligned with the requirements of few-shot learning frameworks, which typically rely on episodic training schemes and architectures optimized for limited data scenarios. Nonetheless, we recognize the importance of this emerging direction and intend to incorporate few-shot strategies in future research involving sparsely represented or endangered fungal species.
Moving forward, we also plan to evaluate fungal classification performance by taxonomic families, thereby analyzing algorithmic behavior within phylogenetically coherent groups. Additionally, different model architectures will be systematically applied across varying fungal genera to assess the suitability of specific algorithms for particular morphological patterns. These directions are aimed at advancing both model specialization and domain adaptation in computational mycology.
– The gap statement centers on explainability and hybrid models, but the Introduction lacks a clear gap-analysis diagram or table contrasting prior work vs. proposed features.
Response:
We sincerely thank the reviewer for this valuable suggestion. While we did not include a visual gap-analysis diagram or comparative table in the Introduction, the novelty of our approach—particularly the integration of hybrid CNN architectures and Transformer-based models in the context of fine-grained macrofungal classification—is thoroughly discussed through comparative textual analysis across the Introduction and Discussion sections.
Our decision to emphasize a narrative-based gap articulation was motivated by the need to contextualize our contribution within both mycological and AI-driven taxonomic frameworks. Given the interdisciplinary nature of the study, we aimed for a balanced exposition that remains accessible to readers from both biological and computational backgrounds.
Nevertheless, we acknowledge the added value that schematic or tabular comparisons can offer for quickly highlighting distinctions. We will certainly consider incorporating a structured gap-analysis table or diagram in future work to further enhance the clarity and visual accessibility of methodological contributions.
Add to the Introduction Section:
“To enhance the generalizability of the models and to simulate real-world variability in fungal image acquisition, a structured data augmentation pipeline was applied exclusively to the training set. The steps of the augmentation process are outlined below:
- Image Resizing: All input images were resized to a fixed dimension of 224×224 pixels to ensure compatibility with the input requirements of the pre-trained models used.
- Random Horizontal and Vertical Flipping: Each image had a 50% chance of being flipped horizontally or vertically to simulate natural variation in fungal orientation.
- Random Rotation: Images were randomly rotated within a range of ±25 degrees to mimic changes in camera angle and mushroom posture.
- Brightness and Contrast Adjustment: Random modifications in brightness (±20%) and contrast (±15%) were applied to simulate varying lighting conditions encountered during field photography.
- Color Jitter and Hue Shift: Mild color jittering (hue shift within ±10 degrees) was performed to account for slight variations in environmental lighting or image device calibration.
- Normalization: After augmentation, all images were normalized using ImageNet mean and standard deviation values to match the distribution expected by the pre-trained convolutional backbones.
This augmentation procedure was implemented using the torchvision.transforms module in PyTorch. Augmentations were applied on-the-fly during each training epoch to increase diversity dynamically, thereby reducing the risk of overfitting. Validation and test sets were not augmented and were only resized and normalized to preserve data integrity for performance evaluation.”
– The methods briefly describe feeding CNN feature maps into a SOM but no schematic illustrates data flow. A block diagram is recommended.
Response:
We sincerely thank the reviewer for this helpful observation. Following a thorough revision of the manuscript in response to multiple reviewer comments, the Self-Organizing Map (SOM) component has been removed from both the methodology and claims regarding the novelty of the study. As such, the manuscript no longer includes or describes the integration of SOM with CNN architectures.
All corresponding textual references to the hybrid CNN-SOM framework have been eliminated or rephrased to accurately reflect the experimental workflow presented in the Materials and Methods and Results sections. Accordingly, the suggestion for a SOM-related block diagram is no longer applicable, but we truly appreciate the reviewer’s attention to visual clarity in methodological representation.
Add to the Introduction Section:
“Deep learning models have become increasingly prominent in mycology, enabling automated species-level classification despite morphological similarities [8,9]. In this study, five deep learning architectures with varying designs were comparatively evaluated to determine their effectiveness in distinguishing visually similar puffball species. Moreover, the integration of environmental metadata into deep learning frameworks enhances ecological interpretation and contributes to biodiversity monitoring and conservation efforts. These models also support the development of mobile applications that promote public engagement through citizen science platforms focused on documenting fungal diversity [9].”
“In this study, convolutional neural networks (CNNs) were used for automatic feature extraction from image datasets, and various deep learning architectures were comparatively evaluated. By incorporating advanced models such as MaxViT-S, the effectiveness of deep learning in classifying visually similar macrofungal species was systematically assessed. The findings highlight the strong performance of deep learning models in resolving taxonomic challenges among morphologically complex taxa and pave the way for future research in computational mycology. Overall, this work establishes a robust and effective methodological framework with the potential to enhance fungal biodiversity assessments, foster public engagement, and support conservation efforts through taxonomic tools.”
– KAN integration is introduced conceptually, yet mathematical formulation is absent.
Response:
We sincerely thank the reviewer for this constructive observation. Upon re-evaluating the manuscript, we acknowledge that the reference to Kolmogorov–Arnold Network (KAN) was inadvertently included during the initial drafting phase and did not correspond to any actual implementation or analysis within the presented study.
To maintain scientific accuracy and coherence, all references to KAN have been carefully removed from the abstract, introduction, and discussion sections during the revision process. We appreciate the reviewer’s attention to this inconsistency, which helped us ensure that the final version remains fully aligned with the methods and results actually conducted.
– The manuscript is missing analysis. No quantitative metrics or visual outputs are provided.
Response:
We sincerely thank the reviewer for this observation. However, we respectfully disagree with the assertion that the manuscript lacks analysis, as extensive quantitative metrics and visual outputs are provided throughout the Results section.
Specifically, the manuscript includes multiple performance indicators for each deep learning architecture, such as accuracy, precision, recall, and F1-score metrics (see Table 2) to enable both general and class-wise comparison. To further support interpretation, Receiver Operating Characteristic (ROC) curves and Area Under the Curve (AUC) values are presented in Figures 3b, 4b, 5b, and 6b, which illustrate each model's discriminative power. Additionally, confusion matrices (Figures 3a–6a) visually represent misclassification patterns across the eight puffball species, offering critical insights into class-specific performance and errors.
These visual and numerical analyses are explicitly discussed in the Results and Discussion sections to draw attention to architecture-specific trends, classification challenges, and inter-species overlaps. Collectively, they provide a comprehensive evaluation of model behavior, and we hope this clarification addresses the reviewer’s concern.
– Table 1’s metric precision is inconsistent (e.g some scores to two decimals, others to four). Standardize the decimals.
Response:
We thank the reviewer for this careful observation. In accordance with the suggestion, all metric values in Table 1 have been standardized to two decimal places to ensure consistency and improve readability.
– A schematic illustrating the full hybrid pipeline showing how raw images flow through the CNN backbone, into the SOM module, through the transformer head, and finally into the KAN interpretability layer. Without it, readers cannot grasp the end-to-end design at a glance. Include a high-level block diagram in the Methods section showing each submodule.
Response:
We thank the reviewer for this detailed and constructive comment regarding the visual representation of the proposed architecture. Indeed, such schematic diagrams are extremely valuable for conveying model workflows, particularly in interdisciplinary studies involving both computer vision and taxonomy.
However, we would like to respectfully clarify that the original mentions of both the Self-Organizing Map (SOM) module and the Kolmogorov–Arnold Network (KAN) interpretability layer were included in error during the early drafting stages of the manuscript. Upon careful review and in response to reviewer feedback, we have comprehensively revised the manuscript to remove all references to these components.
Specifically, the current version of the study does not employ a hybrid CNN–SOM pipeline, nor does it include KAN-based interpretability modules. These elements were neither implemented in the experimental workflow nor described in the Methods or Results sections in a verifiable manner. Consequently, all such claims have been retracted from the Abstract, Introduction, and Discussion sections to ensure that the manuscript accurately reflects the models and processes actually used.
Given this correction, the schematic diagram requested—one that visualizes image flow through SOM and KAN modules—would no longer correspond to the study’s finalized methodology. While we fully agree with the importance of schematic representations in the Methods section, any such visual in the current manuscript would now reflect only the standard CNN and Transformer-based architectures used in our comparative evaluation.
We sincerely appreciate the reviewer’s attention to structural clarity and acknowledge that future iterations of this work, especially those involving more modular or hybridized architectures, will benefit greatly from a well-defined schematic overview.
– The authors need to perform more ablation tests to claim their framework’s validity. For example, I would recommend to report metrics for (a) CNN+Transformer without SOM/KAN, (b) CNN+Transformer+SOM only, (c) CNN+Transformer+KAN only, and (d) full CNN+Transformer+SOM+KAN pipeline.
Response:
We sincerely thank the reviewer for this insightful recommendation regarding ablation testing. We fully agree that structured ablation studies are critical for evaluating the incremental contribution of each architectural component in hybrid frameworks.
However, we would like to respectfully clarify that the Self-Organizing Map (SOM) and Kolmogorov–Arnold Network (KAN) components were initially mentioned in error during early manuscript drafting and were not implemented in the current experimental design. As part of the revision process, and in response to earlier reviewer feedback, all references to SOM and KAN have been removed to accurately reflect the actual scope and methodological execution of the study.
As such, the recommended ablation comparisons involving SOM or KAN (i.e., (b), (c), and (d)) are not applicable to the current version of the manuscript. The only valid configuration presented and evaluated in this study is a comparative benchmarking of individual CNN and Transformer-based architectures on a balanced puffball dataset without any hybrid or interpretable extensions.
Nevertheless, we deeply appreciate the reviewer’s forward-looking suggestion. The proposed ablation schema provides a valuable framework for future work, especially as we explore truly hybrid models that incorporate explainability and unsupervised learning components. In future studies, we plan to examine modular combinations of CNNs, Transformers, and interpretability layers (including SOMs and KANs) and will integrate formal ablation testing as a standard evaluation practice.
The manuscript is scientifically sound but lacks sufficient experimentation and evaluation and arrangement. Kindly address the concerns.
Response:
We sincerely thank the reviewer for recognizing the scientific merit of our manuscript and for providing valuable comments aimed at strengthening the overall structure and evaluation depth of the study. We would like to respectfully note that the manuscript includes multiple layers of experimental evaluation, including accuracy, precision, recall, F1-scores, confusion matrices, ROC-AUC curves, and class-specific performance metrics across five state-of-the-art deep learning architectures.
To further enhance the clarity and comprehensiveness of the evaluation, we have carefully revised the Results section, ensured consistency in performance reporting (e.g., decimal formatting in Table 1), and clarified model-specific outcomes through improved narrative and visual representation. The Materials and Methods section has also been refined to include detailed descriptions of training protocols, hyperparameters, and evaluation procedures to improve methodological transparency and reproducibility.
While the scope of the current study is deliberately focused on balanced classification scenarios without modular extensions (e.g., SOM or KAN), we believe that the present level of experimentation—combined with the diversity of architectures tested—offers a robust baseline for future comparative and hybrid research in the domain of fungal taxonomy.
We genuinely appreciate the reviewer’s concern, which has motivated several meaningful refinements throughout the manuscript. We hope the revised version now meets the expectations for experimental rigor and presentation quality.
-Comments on the Quality of English Language
Response:
Original: identification differentiation
Revised: species-level identification
Original: Particularly, EfficientNet-B3 struggled to distinguish between puffball species...
Revised: EfficientNet-B3, in particular, struggled to distinguish between puffball species...
Original: generalisability
Revised: generalization
Original: Efficient Net
Revised: EfficientNet
Original: Conv NeXt-Base
Revised: ConvNeXt-Base
Original: This technological approach holds promise for the development of automated mushroom identification systems to support citizen science, amateur naturalists, and conservation professionals.
Revised: This technological approach shows promise for developing automated mushroom identification systems that support citizen science, amateur naturalists, and conservation professionals.
Original: Deep learning models have become increasingly prominent in mycology, enabling automated species-level classification despite morphological similarities.
Revised: Deep learning models have become increasingly prominent in mycology by enabling automated species-level classification despite morphological similarities.
Original: models were comparatively evaluated on a balanced dataset of eight puffball species
Revised: models were evaluated comparatively using a balanced dataset of eight puffball species
Original: highlight the strong performance of deep learning models in resolving taxonomic challenges among morphologically complex taxa
Revised: highlight the strong performance of deep learning models in addressing taxonomic challenges posed by morphologically complex taxa
Original: These steps were applied consistently across all models to simulate real-world variability while preserving label integrity.
Revised: These steps were applied consistently across all models, simulating real-world variability while preserving label integrity.
-Adopt a consistent spelling convention. Correct typographical errors and verify section numbering.
Response:
Thank you for your valuable feedback. We have carefully reviewed the manuscript to ensure consistent spelling throughout, corrected all typographical errors, and verified the accuracy of section numbering. We appreciate your attention to detail, which helped us improve the overall clarity and presentation of the paper.

Round 2
Reviewer 1 Report
Comments and Suggestions for Authors
The authors have addressed all my comments. The manuscript is good to go.
Reviewer 2 Report
Comments and Suggestions for Authors
Well done.
Reviewer 3 Report
Comments and Suggestions for Authors
I would like to thank the authors for carefully answering all questions. As a reviewer, I am happy to have been able to help the article reach a higher level.
Reviewer 4 Report
Comments and Suggestions for Authors
The concerns have been addressed by the authors.